# MANY-FOR-MANY: UNIFY THE TRAINING OF MULTIPLE VIDEO AND IMAGE GENERATION AND MANIPULATION TASKS

**Ruibin Li**[1,2]* **Tao Yang**[2]*† **Yangming Shi**[2] **Weiguo Feng**[2]
**Shilei Wen**[2] **Bingyue Peng**[2] **Lei Zhang**[1]†
[1]The Hong Kong Polytechnic University  [2]ByteDance

## ABSTRACT

Diffusion models have shown impressive performance in many visual generation and manipulation tasks. Many existing methods focus on training a model for a specific task, especially, text-to-video (T2V) generation, while many other works focus on finetuning the pretrained T2V model for image-to-video (I2V), video-to-video (V2V), image and video manipulation tasks, *etc*. However, training a strong T2V foundation model requires a large amount of high-quality annotations, which is very costly. In addition, many existing models can perform only one or several tasks. In this work, we introduce a unified framework, namely *many-for-many*, which leverages the available training data from many different visual generation and manipulation tasks to train a single model for those different tasks. Specifically, we design a lightweight adapter to unify the different conditions in different tasks, then employ a joint image-video learning strategy to progressively train the model from scratch. Our joint learning leads to a unified visual generation and manipulation model with improved video generation performance. In addition, we introduce depth maps as a condition to help our model better perceive the 3D space in visual generation. Two versions of our model are trained with different model sizes (8B and 2B), each of which can perform more than 10 different tasks. In particular, our 8B model demonstrates highly competitive performance in video generation tasks compared to open-source and even commercial engines. Our models and source codes are available at https://github.com/leeruibin/MfM.git.

## 1 INTRODUCTION

Visual data generation has a wide range of applications in industry and our daily lives, such as video games (Valevski et al., 2024), advertising (Zhang et al., 2024), media content creation (Polyak et al., 2025), *etc*. Along with the great success of text-to-image (T2I) generation models (Ramesh et al., 2021; Rombach et al., 2021; Podell et al., 2023; Esser et al., 2024), video generation techniques (OpenAI, 2024; Yang et al., 2024c; Polyak et al., 2025; Ma et al., 2025; Kong et al., 2024; Chen et al., 2025; Team, 2025) have recently witnessed significant progress driven by the rapid development of diffusion models (DMs) (Ho et al., 2020; Rombach et al., 2021; Peebles & Xie, 2022; Lipman et al., 2023). Current research is preliminarily focused on text-to-video (T2V) generation. Early attempts (Guo et al., 2023; Blattmann et al., 2023b;a) are often built on pre-trained T2I models such as Stable Diffusion (SD) (Rombach et al., 2021) by encoding motion dynamics into latent codes (Khachatryan et al., 2023) or inserting additional temporal layers (Guo et al., 2023; Blattmann et al., 2023b;a). Despite significant advancements, these methods tend to produce unnatural motions and are limited by the small number of generated frames.

Recently, diffusion transformers (DiT) (Peebles & Xie, 2022; Esser et al., 2024; Yang et al., 2024c) have been widely adopted in numerous image and video generation methods (Esser et al., 2024; Labs, 2024; OpenAI, 2024; Team, 2025) due to their excellent scalability. In particular, SORA (OpenAI, 2024) demonstrates remarkable performance in creating realistic videos, inspiring many subsequent

---

*Equal contribution.
†Corresponding author.

Table 1: The size and supported tasks of the current main open-source video foundation models.

| Model | Size | Training Data | | Supported Tasks | Unified Training |
|---|---|---|---|---|---|
| | | Video | Image | | |
| CogVideoX (Yang et al., 2024c) | 28&5B | unkown | unkown | T2V, I2V | ✗ |
| MovieGen (Polyak et al., 2025) | 30B | 100M | 1B | T2V, Peronalized T2V (PT2V) | ✗ |
| StepVideo (Ma et al., 2025) | 30B | 2B | 3.8B | T2V, I2V | ✗ |
| HunyuanVideo (Kong et al., 2024) | 13B | $\mathcal{O}(100)$M | $\mathcal{O}(1)$B | T2V, I2V | ✗ |
| Wan2.1 (Team, 2025) | 1.3B&14B | 1.5B | 10B | T2V, I2V | ✗ |
| MfM | 2B&8B | 120M | 160M | T2V, I2V, video extension, FLF2V, FLC2V, video manipulation, etc. | ✓ |

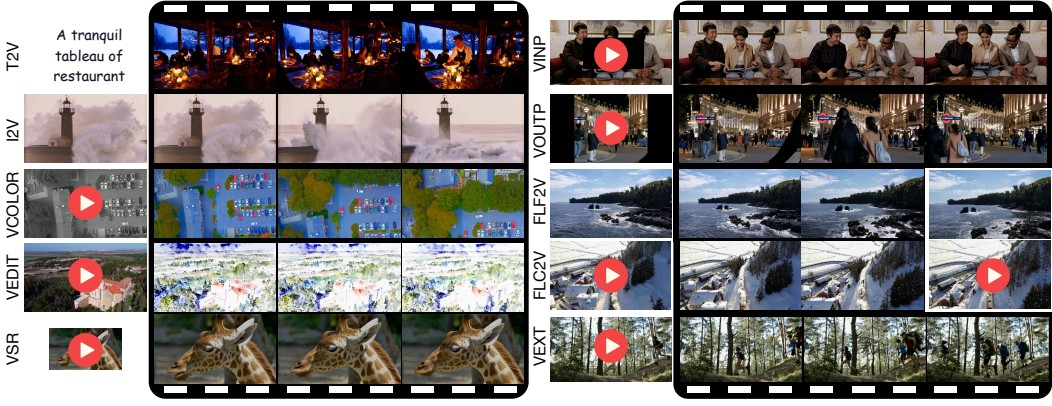

Figure 1: Examples of MfM on typical video generation and manipulation tasks. Generated frames are highlighted within black boxes. Note that MfM uses a single model to perform these tasks.

T2V works (Yang et al., 2024c; Ma et al., 2025; Kong et al., 2024; Team, 2025; RunwayML, 2023; Kuaishou, 2024). For example, trained on web-scale datasets, the open-source models CogVideoX (Yang et al., 2024c), HunyuanVideo (Kong et al., 2024) and Wan2.1 (Team, 2025) have attracted significant attention. The commercial models Runway (RunwayML, 2023) and Kling (Kuaishou, 2024) have demonstrated impressive performance in practical use. Other video generation tasks, such as image-to-video (I2V) and video-to-video (V2V), are commonly regarded as downstream problems of T2V. By fine-tuning pre-trained T2V models with relatively small resources (Yang et al., 2024c; Kong et al., 2024; Team, 2025), various models tailored to different tasks can be obtained, including I2V (Tian et al., 2025; Team, 2025), video super-resolution (Xie et al., 2025), reference-to-video (Liu et al., 2025; Jiang et al., 2025), *etc*.

In this work, we aim to economically train a model from scratch, which can, however, perform a number of visual generation and manipulation tasks effectively, including T2V, I2V, V2V, *etc*. To this end, we introduce a simple yet effective framework, called *Many-for-Many* (MfM in short), to unify the training of different tasks. The key difference between the various visual generation/manipulation tasks lies in their varying conditions. Therefore, we propose to standardize the conditions using a lightweight adapter, thereby enabling multi-task joint training. Adhering to the foundation model's training recipe, we progressively update our MfM model from a low resolution to higher resolutions. Specifically, we employ a joint image-video learning strategy, which equips our model with capabilities for both image generation and manipulation. An advantage of our MfM training framework is that the many data that cannot be used to train T2V models in previous methods now can be used to train our unified model. Therefore, MfM learning not only leads to a unified model but also enhances video generation performance.

Two versions (2B and 8B) of our MfM model are trained. As shown in Table 1, our model can perform more than 10 different visual generation and manipulation tasks. Figure 1 illustrates some examples of MfM tasks. Extensive experiments are performed to demonstrate the effectiveness and flexibility of our MfM model. In particular, our 8B model achieves highly competitive performance in the challenging T2V and I2V tasks by using only 10% of the training data used in state-of-the-art open-source T2V foundation models (Yang et al., 2024c; Kong et al., 2024; Team, 2025).

## 2 RELATED WORK

**Diffusion Models for Visual Generation**. Since the seminal work of denoising diffusion probabilistic model (DDPM) (Ho et al., 2020), remarkable progress has been achieved in training diffusion models (DMs) for image/video generation (Rombach et al., 2021; Esser et al., 2024; Blattmann et al., 2023a; OpenAI, 2024; Kong et al., 2024). In particular, Rombach *et al.* (Rombach et al., 2021) proposed to train DMs in latent space, achieving impressive image generation results with reduced computational costs. The development of Stable Diffusion (SD) (Rombach et al., 2021) has sparked a surge of research in text-to-image (T2I) generation (Podell et al., 2023; Zhang & Agrawala, 2023; Ruiz et al., 2023). SDXL (Podell et al., 2023) expands SD by using a larger model and more sophisticated architecture design. With the advancement in Diffusion Transformer (DiT) (Peebles & Xie, 2022) and Flow Matching (FM) (Lipman et al., 2023), Esser et al. (Esser et al., 2024) proposed MMDiT to train SD3 and Flux (Labs, 2024), which show state-of-the-art T2I performance.

In terms of T2V generation, early efforts often fine-tune pre-trained T2I models to learn motion dynamics (Guo et al., 2023; Blattmann et al., 2023a; Chen et al., 2024a), which are, however, limited in motion naturalness and video frames. The success of SORA (OpenAI, 2024) in generating realistic long videos has inspired numerous commercial (OpenAI, 2024; RunwayML, 2023; Kuaishou, 2024) and open-source (Yang et al., 2024c; Polyak et al., 2025; Ma et al., 2025; Kong et al., 2024; Team, 2025) T2V models. CogVideoX (Yang et al., 2024c) adopts MMDiT to T2V and achieves impressive results in modeling coherent long-duration videos with natural movements. Ma *et al.* (Ma et al., 2025) and Polyak *et al.* (Polyak et al., 2025) scaled the T2V model to $30B$ and demonstrated promising improvements in simulating natural motions. Specifically, Ma *et al.* (Ma et al., 2025) employed video-based direct preference optimization (Rafailov et al., 2024), namely Video-DPO, to improve the visual quality of generated videos. The recently released open-source models HunyuanVideo (Kong et al., 2024) and Wan2.1 (Team, 2025) exhibit much improved video quality and prompt controllability, significantly facilitating the research of video generation in the community.

**Downstream Tasks of Visual Generation Models**. With the advancement in pre-trained T2I and T2V foundation models, researchers have developed various techniques to adapt them to various content creation and manipulation tasks, such as controllable generation (Zhang & Agrawala, 2023; Wu et al., 2023), personalized generation (Ruiz et al., 2023), editing (Brooks et al., 2023; Liew et al., 2023), super-resolution (Yang et al., 2023), among others. Zhang *et al.* (Zhang & Agrawala, 2023) introduced ControlNet to facilitate various conditional inputs, which, however, requires multiple control modules for different conditions. UniControl (Qin et al., 2023) and UNIC-Adapter (Duan et al., 2024) enable unified conditional image generation using a single model. InstructPix2Pix (Brooks et al., 2023) and MagicBrush (Zhang et al., 2023a) offer general-purpose image editing solutions. However, for video tasks, most approaches (Wu et al., 2023; Liew et al., 2023) still follow a single-model single-task framework due to the complexities of video generation. Very recently, Jiang *et al.* (Jiang et al., 2025) proposed a so-called all-in-one model for multiple visual creation and editing tasks based on pre-trained T2V models (Kong et al., 2024; Team, 2025). Although achieving impressive results, this model is built on pre-train T2V models and treats the other tasks as downstream applications. In contrast, in this work, we train a single model from scratch, which can, however, perform multiple visual generation and manipulation tasks, by effectively utilizing the available training data from different tasks.

## 3 MANY-FOR-MANY UNIFIED TRAINING

Our Many-for-Many (MfM) unified training framework is illustrated in Figure 2. It is basically a DiT (Peebles & Xie, 2022) with 3D full attention, trained by the Flow Matching technique (Lipman et al., 2023). Videos and text prompts are encoded using a video VAE (Yang et al., 2024c) and an LLM text encoder (Raffel et al., 2020), respectively. To mitigate the reliance on costly annotation of T2V training data and make the best use of existing training data from various visual generation and manipulation tasks, we introduce an effective and lightweight adapter to unify the various conditions across different tasks. A progressive and joint training strategy is then developed to train a unified model for multiple visual generation and manipulation tasks. To accommodate varying computational demands and performance requirements, we design two versions of our model with different sizes (8B and 2B), whose hyper-parameters are summarized in Table 7 of the **Appendix**. The inference latency of different variants on different resolution are summarized in Table 8 of the **Appendix**.

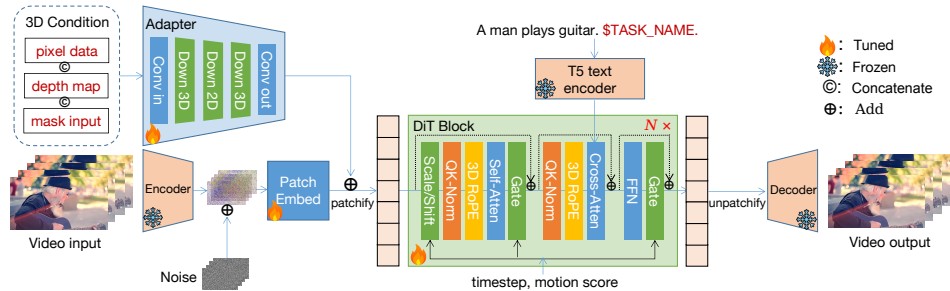

Figure 2: Architecture of the proposed Many-for-Many (MfM) unified training framework.

## 3.1 ADAPTER FOR DIFFERENT INPUTS

We categorize the inputs of different image and video tasks (OpenAI, 2024; Hu et al., 2023; Wu et al., 2023; Jiang et al., 2025) based on their dimensions: 0D conditions (*e.g.*, timestep and motion score), 1D conditions (*e.g.*, text), 2D conditions (*e.g.*, image and mask) and 3D conditions (*e.g.*, video and video depths). 0D and 1D conditions are commonly used in DiT, which are embedded using AdaLN and a text encoder, respectively. 2D conditions can be padded to 3D and thus merged into 3D conditions. The 3D conditions include both pixel data (*e.g.*, image and video) and masks, which vary across generation and manipulation (including enhancement) tasks:

- **Generation Tasks**: These tasks require at least one frame to be generated without any frame-wise condition. Examples include T2I (text-to-image), T2V (text-to-video), I2V (image-to-video), video extension, FLF2V (first-last-frame-to-video) and FLC2V (first-last-clip-to-video).

- **Manipulation Tasks**: These tasks require frame-wise conditions. Examples include image/video inpainting/outpainting, image/video colorization, image/video style transfer, single image super-resolution (SISR), video super-resolution (VSR), *etc*.

As illustrated in the upper-left corner of Figure 2, fortunately, we can represent the various inputs in a unified manner, concatenating the pixel, depth map and mask conditions. The depth maps are introduced as a condition to enhance our model's understanding of 3D space. Note that we append the task name (*e.g.*, "text-to-video", "image-to-video", *etc.*) to the text prompts to clarify tasks because some of them share a common video mask input, such as VSR and video colorization. Figure 3 illustrates some example inputs for different generation and manipulation tasks. For instance, for the T2V task, the pixel data, depth map, and mask inputs are all set to 0 so that the task is driven by merely the text prompt. For the task of I2V, only the conditions of the first frame are provided.

Existing visual generation methods typically process the pixel and mask conditions separately — pixel conditions are processed by video VAE, while mask conditions are directly reshaped and interpolated (Jiang et al., 2025; Team, 2025). While achieving impressive results, these methods are complex and cannot be easily extended to other types of conditions such as depth maps. Our proposed adapter unifies all 3D inputs, regardless of their content (*e.g.*, pixel, mask, depth). The adapter comprises several convolution layers and downsampling blocks to adjust the temporal and spatial resolutions. Given a 3D condition input in pixel space $Y \in \mathbb{R}^{T \times H \times W \times C}$, where $\{T, H, W, C\}$ represents the frame number, height, width, and channel number, the adapter converts it into a feature map $y \in \mathbb{R}^{t \times h \times w \times c}$, which shares the same spatial and temporal resolution as the latent space of the video VAE and is added to it. Given the video VAE's $8 \times 8$ spatial and $4\times$ temporal compression ratios (Yang et al., 2024c), we have $t = T/4, h = H/8, w = W/8$. The proposed architecture can be easily adjusted according to the compression ratios of alternative video VAEs.

## 3.2 TRANSFORMER WITH 3D FULL ATTENTION

**3D Full Attention**. Early video generation models (Blattmann et al., 2023a; Guo et al., 2023) are typically built on pre-trained T2I models, which use separate spatial and temporal attention to reduce computational complexity. However, such methods are suboptimal for modeling natural motions (Yang et al., 2024c). In recent works (OpenAI, 2024; Yang et al., 2024c; Ma et al., 2025; Kong et al., 2024), 3D full attention has become widely adopted and shown superiority in generating videos

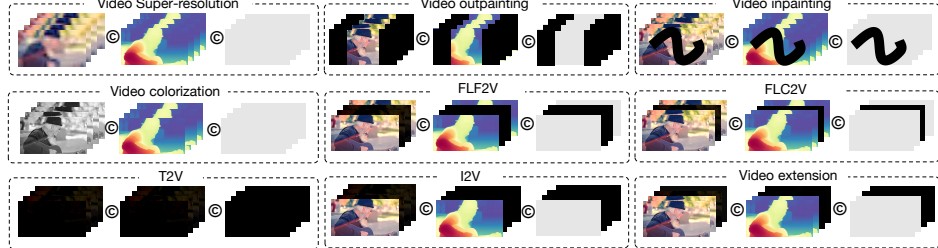

Figure 3: Example conditional inputs for some visual generation and manipulation tasks. In each task block, from left to right, the conditions are respectively task-oriented pixel data, depth maps, and mask inputs. The mask is composed of binary black and white pixels, with white pixels indicating the regions that are conditioned on pixel data, and black pixels indicating the regions to be generated.

with smooth and consistent motions. In this work, we incorporate the transformer block in GoKu (Chen et al., 2025), which consists of a self-attention module to capture relationships within input sequences, a cross-attention layer to include text embeddings, and an adaptive layer normalization (AdaLN) operation to embed timestep and motion score information.

**3D RoPE**. Rotary Position Embedding (RoPE) (Su et al., 2021) encodes positional information and enables the model to understand both the absolute position of tokens and their relative distances, demonstrating powerful ability in capturing inter-token relationships, particularly for long sequences in LLMs. We extend it to 3D RoPE by applying 1D RoPE to each temporal ($t$) and spatial ($h, w$) dimension, then concatenating the encodings. Specifically, for 3D video data ($t, h, w$), each dimension occupies $2/8$, $3/8$, and $3/8$ of the hidden state channels, respectively. We apply 3D RoPE for both image and video tokens. Due to the exceptional extrapolation capabilities of RoPE, the proposed 3D RoPE can effectively handle videos with varying resolutions and lengths.

**Q-K Normalization**. Previous methods (Esser et al., 2024; Dehghani et al., 2023a) have shown that the training of large transformer models can encounter numerical instability due to the uncontrollable growth in attention entropy. To address this issue, following (Esser et al., 2024; Dehghani et al., 2023a), we adopt RMSNorm (Zhang & Sennrich, 2019) and implement Query-Key Normalization (QK-Norm) to stabilize the training process.

## 3.3 TRAINING DETAILS

**Flow Matching**. During model training, we employ Rectified Flow (RF) to optimize the network due to its superior performance (Lipman et al., 2023; Esser et al., 2024; Chen et al., 2025). In each training step, a video input $X_0$, Gaussian noise $\epsilon \sim \mathcal{N}(0, 1)$, and a timestep $t \in [0, 1]$ are randomly sampled. The model input $X_t$ is calculated as a linear interpolation between $\epsilon$ and $X_0$:

$$X_t = (1 - t)X_0 + t\epsilon. \tag{1}$$

The model is trained to approximate the ground-truth velocity $V_t = \frac{dX_t}{dt} = \epsilon - X_0$, which represents the change rate of $X_t$ with respect to timestep $t$, capturing the change direction and magnitude from $\epsilon$ to $X_0$. Given conditions of motion score $ms$, text prompt $c$, and 3D conditional input $Y$, we train our model $\mu_\theta$ to predict the velocity $V_t$. The optimization objective $\mathcal{L}$ is defined as:

$$\mathcal{L} = \mathbb{E}_{t, X_0, \epsilon \sim \mathcal{N}(0,1), ms, c, Y} |\mu_\theta(t, X_t, ms, c, Y) - V_t|^2. \tag{2}$$

Following SD3 (Esser et al., 2024), we use Logit-Normal Sampling in training.

**Multi-Task Joint Learning**. While our model is primarily designed for video generation, we leverage a large volume of image data in training. Following existing T2V foundation models (OpenAI, 2024; Kong et al., 2024; Chen et al., 2025), we progressively adjust the image-to-video ratio throughout training. Initially, we train with pure text-image pairs to establish a connection between textual prompts and high-level visual semantics. As training progresses, we inject video data, gradually decreasing the image-to-video ratio to 0.1. This image-video joint learning strategy expands our training data and enables our model to tackle various image tasks, including T2I and SISR.

Unlike standard T2V foundation models, our training data include a substantial portion of low-resolution, watermarked, text-dominated, and concisely captioned data. To effectively utilize the

Table 2: Resolution progressive training recipe for 8B MfM.

| Training Stage | Dataset | SP | bs/GPU | Learning rate | #iters | #seen samples |
|---|---|---|---|---|---|---|
| 128px | 160M images 120M videos | 1 | 16 | 1e-4 | 170k | 700M |
| 360px | 160M images 120M videos | 1 | 2 | 8e-5 | 100k | 100M |
| 720px | 160M images 10M videos | 2 | 1 | 5e-5 | 50k | 12M |
| Multi-res | 160M images 5M videos | 2 | 1 | 5e-5 | 40k | 5M |

available training data, we implement multi-task learning, thanks to our proposed conditional adapter. At each training step, we randomly sample a video input, assign a set of qualified tasks that fits it, and select one task to construct the conditional input for training. For each qualified task set, the selection probability of tasks like T2I, T2V, and I2V is tripled compared to other tasks, ensuring that the learning process pay more attention to more challenging problems.

**Resolution Progressive Training**. Our training pipeline is structured into multiple stages with progressively increased spatial and temporal resolutions. Initially, we train our model on low-resolution data (*e.g.*, $49 \times 128 \times 224$) at a low computational cost. We then increase the resolution to $89 \times 352 \times 640$ to enhance the model's fine-grained understanding of text-motion relationships. Subsequently, the training resolution is increased to $97 \times 720 \times 1280$ to capture intricate details. Finally, we conclude the training pipeline with a multi-resolution stage using NaViT (Dehghani et al., 2023b). In this stage, the model is fed high-quality videos with their native aspect ratios, dynamically adjusting the durations to limit the total sequence length. This multi-resolution fine-tuning stage enables our model to generate videos at arbitrary resolutions. During training, we randomly replace 10% (30%) of text prompts with null-text prompts for T2V/I. For tasks other than T2V/I, we randomly zero the 3D conditional inputs with a chance of 10%. The detailed training recipe for our 8B model is summarized in Table 2. We adopt Fully Sharded Data Parallelism (FSDP) (Zhao et al., 2023) and Sequence-Parallelism (SP) (Li et al., 2022) to achieve efficient and scalable training of MfM.

## 4 EXPERIMENTS

### 4.1 EXPERIMENT SETUP

**Training Data Preparation**. Our training data are collected from a variety of sources, including publicly available academic datasets, Internet resources, and proprietary datasets. We adopt a data curation pipeline similar to GoKu (Chen et al., 2025) to filter the collected data, obtaining 160M HQ text-image pairs and 40M HQ text-video pairs. We also retrieve 80M relatively LQ text-video pairs for training. We utilize RAFT (Teed & Deng, 2020) to obtain motion scores by computing the mean optical flow of video clips, which are integrated into our MfM model training via AdaLN.

Note that we use significantly fewer text-video pairs than the main T2V models (Polyak et al., 2025; Ma et al., 2025; Kong et al., 2024; Team, 2025) to train our MfM model. However, our MfM framework leverages a multi-task data augmentation strategy to expand the effective training data distribution (please refer to **Appendix** for details), with which we significantly expand the model's exposure to diverse conditioning scenarios without requiring additional data collection. For all tasks, we employ a lightweight depth model (Yang et al., 2024a) to predict the depth maps of the inputs on the fly. We concatenate these depth maps into the 3D conditional inputs as depicted in Figure 3.

**Evaluation**. We utilize the widely used VBench (Huang et al., 2024) to evaluate MfM's performance on T2V and I2V tasks. While a benchmark is proposed in VACE (Jiang et al., 2025) to evaluate a model's multi-task capacity, only one video is open-sourced for each task, and many tasks supported by MfM are not involved in VACE. Therefore, we build an MfM-benchmark, which comprises 480 samples (30 per task) distributed across 16 distinct generation/manipulation tasks (please refer to **Appendix** for details). For all experiments, we maintain the same MfM inference parameters: 30 diffusion steps with a classifier-free guidance scale of 9.0.

Regarding evaluation metrics, on VBench we adopt a comprehensive set of perceptual metrics: aesthetic quality, imaging quality, motion smoothness, dynamic degree, object class accuracy, multiple object handling, spatial relationship preservation, scene consistency, appearance style, temporal style,

Table 3: Quantitative comparison of T2V generation performance on the VBench-T2V benchmark. Comparison baselines are selected from VBench leaderboard. For each dimension, the best result is in bold, the second best result is underscored and the third best result is italic. (Aesth: Aesthetic Quality; Img: Imaging Quality; Mul.Obj: Multiple Objects; Temp: Temporal Style; Consist: Overall Consistency; Avg: Average Ranking.)

| Model | Motion | Dynamic | Aesth. | Img. | Object | Mul.Obj. | Spatial | Scene | Appear. | Temp. | Consist. | Avg. |
|---|---|---|---|---|---|---|---|---|---|---|---|---|
| MfM | 0.983 | _0.819_ | _0.645_ | 0.662 | *0.927* | _0.782_ | _0.802_ | _0.546_ | **0.251** | *0.247* | **0.277** | **2.86** |
| Wan2.1 ((Team, 2025)) | 0.969 | **0.943** | 0.615 | *0.672* | **0.942** | **0.814** | **0.810** | 0.536 | 0.211 | **0.256** | _0.274_ | *3.77* |
| Hunyuan ((Kong et al., 2024)) | *0.989* | 0.708 | 0.603 | _0.675_ | 0.861 | 0.685 | 0.686 | 0.538 | 0.198 | 0.238 | 0.264 | 5.73 |
| Sora ((OpenAI, 2024)) | 0.987 | *0.799* | *0.634* | **0.682** | _0.939_ | *0.708* | 0.742 | **0.569** | _0.247_ | 0.250 | 0.262 | _3.27_ |
| Gen-3 ((RunwayML, 2023)) | _0.992_ | 0.601 | 0.633 | 0.668 | 0.878 | 0.536 | 0.650 | *0.545* | 0.243 | *0.247* | *0.266* | 4.77 |
| PikaLabs ((Labs, 2023)) | **0.995** | 0.475 | 0.620 | 0.618 | 0.887 | 0.430 | 0.610 | 0.498 | 0.222 | 0.242 | 0.259 | 6.95 |
| LTX-Video ((HaCohen et al., 2024)) | *0.989* | 0.543 | 0.598 | 0.602 | 0.834 | 0.454 | 0.654 | 0.510 | 0.214 | 0.226 | 0.251 | 7.82 |
| CogVideoX1.5 ((Yang et al., 2024c)) | 0.981 | 0.561 | 0.620 | 0.653 | 0.834 | 0.672 | *0.794* | 0.532 | *0.246* | _0.254_ | _0.274_ | 5.23 |
| EasyAnimate ((Xu et al., 2024)) | 0.980 | 0.571 | **0.694** | 0.585 | 0.895 | 0.668 | 0.761 | 0.543 | 0.230 | 0.246 | 0.264 | 4.59 |

Table 4: Quantitative comparison of I2V generation performance on the VBench-I2V benchmark. Comparison baselines are selected from VBench leaderboard. For each dimension, the best result is in bold and the second best result is underscored. (IS. Consist: Image Subject Consistency; IB. Consist: Image Background Consistency.)

| Model | IS. Consist. | IB. Consist. | Motion | Dynamic | Aesth. | Img. | Avg. |
|---|---|---|---|---|---|---|---|
| MfM | *0.982* | _0.991_ | *0.987* | *0.613* | 0.608 | **0.718** | **3.33** |
| Wanx-I2V (Team, 2025) | 0.973 | 0.981 | 0.978 | _0.678_ | 0.615 | 0.708 | 5.50 |
| Hunyuan-I2V (Kong et al., 2024) | **0.988** | **0.992** | **0.994** | 0.239 | 0.617 | 0.700 | *3.67* |
| Magi-1 (Sand-AI, 2025) | _0.983_ | 0.990 | 0.986 | **0.682** | *0.647* | 0.697 | _3.50_ |
| Step-Video (Ma et al., 2025) | 0.978 | 0.986 | _0.992_ | 0.487 | 0.622 | 0.704 | 3.83 |
| DynamicCrafter (Xing et al., 2023) | 0.981 | 0.986 | 0.973 | 0.474 | **0.664** | 0.693 | 5.33 |
| VideoCrafter-I2V (Chen et al., 2024a) | 0.911 | 0.913 | 0.980 | 0.226 | 0.607 | _0.716_ | 7.83 |
| I2VGen-XL (Zhang et al., 2023b) | 0.975 | 0.976 | 0.983 | 0.249 | _0.653_ | 0.698 | 5.83 |
| CogvideoX-I2V (Yang et al., 2024c) | 0.971 | 0.967 | 0.984 | 0.331 | 0.618 | 0.700 | 6.17 |
| ConsistI2V (Ren et al., 2024) | 0.958 | 0.959 | 0.973 | 0.186 | 0.590 | 0.669 | 9.50 |

and overall consistency (higher scores indicate better performance across all metrics). Meanwhile, we rank the competitors for each metric and calculate the average rank over all metrics for each method. For some tasks on MfM-benchmark, we also use reference-based metrics, including FID, PSNR, SSIM, and LPIPS, to quantify the fidelity of generated content.

## 4.2 EXPERIMENTAL RESULTS ON T2V AND I2V

Since most of the existing methods use two separate models for T2V and I2V tasks, we present the quantitative comparison in two tables. The results of T2V are shown in Table 3. We can see that MfM achieves the best average rank (2.86) among all models evaluated. In particular, MfM exhibits well-balanced performance across multiple dimensions, ranking the best in appearance and overall consistency, and the second in dynamic degree, aesthetic quality, multiple object generation, and spatial relation generation, which are essential for producing visually coherent videos aligned with textual descriptions. In comparison, the larger models such as Wan2.1 (14B), Hunyuan (13B) and the commercial models such as Sora can achieve impressive scores in specific dimensions, but their overall performance is compromised by notable weaknesses in other dimensions. For instance, Wan2.1 ranks last in motion smoothness, while Hunyuan shows deficiencies in appearance style, resulting in jerky movements, visual distortions, or monotonous video style in some scenarios. The visual comparison can be found in Figure 4, where Wan2.1 generates a bicycle without a rider and fails to depict the slowing motion instruction given in the prompt. Similarly, Sora and Hunyuan fail to accurately represent the slowing motion. Hunyuan also exhibits distortion in the bicycle's handlebars as the sequence progresses. Our MfM successfully generates a motion-consistent video with the bicycle correctly slowing down, demonstrating superior temporal coherency.

The results of I2V are shown in Table 4. We see that MfM also achieves the best average rank (3.33). In particular, it excels in imaging quality and achieves very balanced performance across static consistency and dynamic generation. In comparison, although Hunyuan-I2V achieves the highest scores in consistency and motion smoothness, its performance in dynamic degree and aesthetic qualities is substantially lower, resulting in an average rank of only 3.67, lower than MfM and Magi-1. Visual comparisons of I2V generation are provided in the **Appendix**.

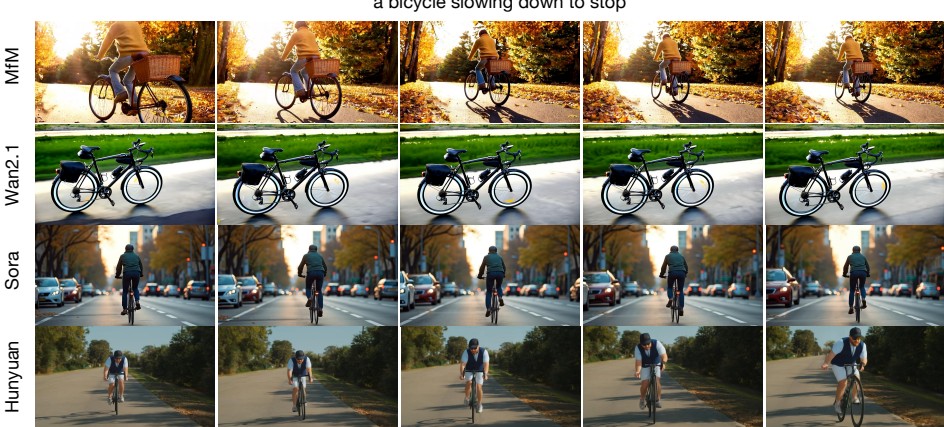

Figure 4: Qualitative comparison of T2V generation results on the prompt "a bicycle slowing down to stop." More visual comparisons are provided in the **Appendix**.

Table 5: Performance comparison on multiple video manipulation tasks.

| Task | Method | Reference-based Metrics | | | | No-reference Perceptual Metrics | | | | |
|---|---|---|---|---|---|---|---|---|---|---|
| | | FID↓ | PSNR↑ | SSIM↑ | LPIPS↓ | Aesth. | Img. | Motion | Consist. | Temp. |
| VINP | MfM | **51.03** | **21.31** | 0.830 | **0.112** | 0.560 | **0.767** | **0.994** | 0.220 | **0.985** |
| | VACE ((Jiang et al., 2025)) | 67.51 | 17.41 | 0.534 | 0.256 | **0.569** | 0.757 | 0.990 | **0.224** | 0.983 |
| | ProPainter ((Zhou et al., 2023)) | 119.93 | 20.40 | **0.880** | 0.118 | 0.417 | 0.739 | 0.992 | 0.206 | **0.985** |
| VOUTP | MfM | **44.15** | **18.21** | **0.733** | **0.168** | 0.539 | **0.745** | **0.992** | **0.216** | **0.974** |
| | VACE ((Jiang et al., 2025)) | 54.34 | 16.16 | 0.500 | 0.310 | **0.567** | 0.736 | 0.987 | 0.211 | 0.971 |
| | FYC ((Chen et al., 2024b)) | 94.69 | 14.49 | 0.416 | 0.414 | 0.550 | 0.736 | 0.988 | 0.211 | 0.971 |
| | M3DDM ((Fan et al., 2023)) | 174.61 | 17.96 | 0.571 | 0.475 | 0.484 | 0.671 | 0.982 | 0.214 | 0.972 |
| FLF2V | MfM | **31.98** | **19.95** | **0.583** | **0.203** | **0.525** | 0.730 | 0.981 | 0.225 | 0.966 |
| | Wanx ((Team, 2025)) | 38.24 | 18.28 | 0.512 | 0.244 | 0.520 | **0.742** | **0.990** | **0.229** | **0.978** |
| | Hunyuan ((Kong et al., 2024)) | 118.18 | 10.17 | 0.372 | 0.419 | 0.476 | 0.598 | 0.992 | 0.225 | 0.985 |
| VCOLOR | MfM | **76.54** | 17.93 | 0.810 | 0.176 | 0.582 | 0.756 | **0.993** | **0.230** | **0.985** |
| | colormnet ((Yang et al., 2024b)) | 77.08 | 17.47 | **0.812** | **0.160** | **0.594** | **0.758** | 0.990 | **0.230** | 0.980 |
| | TCVC ((Zhang et al., 2023c)) | 82.42 | **20.69** | 0.699 | 0.201 | 0.553 | 0.720 | 0.991 | 0.228 | 0.984 |

Finally, it is worth mentioning that our MfM achieves competitive results in both T2V and I2V generation tasks using a **single unified model**, while previous approaches such as Wan and Hunyuan rely on separate specialized models for each generation paradigm. The unified nature of MfM reduces overall model parameters and ensures consistent visual quality between text and image conditioning.

## 4.3 PERFORMANCE ON MULTIPLE VIDEO MANIPULATION TASKS

Beyond T2V and I2V generation, our MfM supports 16 distinct tasks through a unified model. Given that our primary focus is on video generation and manipulation, while the image tasks and data are used to aid video task training, we perform evaluation on a subset of video tasks with established baselines for comparison. Specifically, we select four representative tasks, including video inpainting, video outpainting, video transition and video colorization, for experiment since they have competitive baseline models and standardized evaluation protocols. Table 5 presents quantitative comparisons on our established MfM-Benchmark. Visual comparisons can be found in the **Appendix**.

Our experimental results demonstrate MfM's excellent versatility and effectiveness as a unified video foundation model across diverse manipulation tasks. First, MfM shows consistent advantages in reference-based metrics. In particular, it achieves FID improvements ranging from 16.4% to 72.9% over the specialized models of these tasks. In addition to reference-based metrics, MfM exhibits impressive temporal coherence, which demonstrates MfM's strong ability to seamlessly transition between different operations: inferring complex motion change from two frames, preserving spatial coherence during region manipulation, and maintaining consistent appearance while modifying visual attributes. Meanwhile, with MfM the knowledge learned from one task can benefit another task. For example, the capability developed for handling boundaries in outpainting can enhance performance in inpainting; similarly, the motion-inference ability required for video translation contributes to the temporal coherence observed in colorization tasks. In summary, MfM can effectively capture the principles underlying diverse video manipulation tasks and achieve competitive performance without requiring separate architectures for each manipulation paradigm.

Table 6: Ablation study on multi-task training versus single-task training on VBench-T2V. The best result is in **bold**, the second best result is underscored. Meanwhile, green box means better than T2V training paradigm while blue box means worse than T2V training paradigm.

| Paradigm | Motion | Dynamic | Aesth. | Img. | Object | Mul.Obj. | Spatial | Scene | Appear. | Temp. | Consist. |
|---|---|---|---|---|---|---|---|---|---|---|---|
| T2V | 0.987 | 0.806 | 0.617 | 0.599 | 0.926 | 0.705 | 0.638 | 0.502 | 0.232 | 0.249 | 0.263 |
| T2V+I2V | 0.988 | 0.778 | 0.612 | 0.601 | 0.918 | 0.665 | 0.611 | 0.533 | 0.236 | 0.248 | 0.261 |
| T2V+VCOLOR | 0.987 | 0.764 | 0.621 | 0.609 | 0.948 | 0.723 | 0.593 | 0.543 | 0.233 | 0.251 | 0.265 |
| T2V+VSR | 0.989 | 0.792 | 0.621 | 0.597 | 0.930 | 0.664 | 0.608 | 0.555 | 0.230 | 0.252 | 0.261 |
| T2V+VINP | **0.990** | 0.778 | 0.623 | 0.610 | 0.945 | **0.767** | 0.652 | 0.537 | 0.229 | 0.247 | 0.265 |
| T2V+VOUT | 0.988 | 0.722 | 0.625 | 0.614 | 0.948 | 0.753 | 0.612 | 0.520 | 0.229 | 0.250 | 0.261 |
| T2V+FLF2V | 0.985 | 0.847 | 0.620 | 0.613 | **0.960** | 0.658 | 0.619 | 0.531 | 0.233 | 0.252 | 0.262 |
| T2V+FLC2V | 0.984 | 0.875 | 0.615 | 0.598 | 0.919 | 0.695 | 0.663 | **0.568** | 0.231 | 0.250 | 0.262 |
| T2V+VEXT | 0.988 | 0.722 | **0.626** | **0.615** | 0.941 | 0.761 | 0.621 | 0.563 | 0.235 | 0.251 | 0.263 |
| MfM w/o Depth | 0.988 | 0.819 | 0.623 | 0.584 | 0.908 | 0.659 | 0.675 | 0.472 | 0.232 | 0.248 | 0.264 |
| MfM w/ Depth | 0.988 | **0.903** | 0.625 | 0.608 | 0.953 | 0.723 | **0.677** | 0.536 | **0.237** | **0.253** | **0.266** |

## 4.4 THE BENEFIT OF MULTI-TASK TRAINING TO VIDEO GENERATION

We conduct a series of ablation studies to validate that our multi-task training strategy benefits video generation, and to examine the influence of different auxiliary tasks and our design choices. Specifically, initialized from a T2V baseline model, we train models under several settings with the same number of training iterations, including: (i) training with the pure T2V paradigm, (ii) T2V augmented with a single auxiliary task, (iii) our final MfM, and (iv) MfM without depth conditioning. The results on the VBench-T2V benchmark are reported in Table 6.

First, we see that all variants outperform the baseline on Scene metrics, validating the value of adding auxiliary tasks on scene detail generation. Beyond this, different tasks can yield distinct gains. For example, VINP and VOUT boost semantic metrics (e.g., Object, Multi-Object). In contrast, VEXT improves perceptual quality (Aesthetic, Imaging Quality). Interestingly, most variants degrade Dynamics, whereas FLF2V and FLC2V improve it. This is because both of them require interpolating realistic motion between states, providing temporally grounded, geometrically constrained signals to supervise the model to learn motion realism, temporal consistency, and dynamic integrity, resulting in smoother, more structured temporal dynamics.

Second, MfM consistently outperforms the pure T2V baseline across all metrics. This is because compared with T2V that relies solely on high-level text signal, MfM leverages complementary low- and mid-level signals (e.g., color stability, spatial completion, motion plausibility). This multi-task synergy improves not only semantic alignment, but also detail coherence and visual realism. Moreover, MfM adaptively balances task-specific inductive biases, avoiding overfit to any single objective. We attribute this to multi-task regularization: diverse supervisory signals encourage the model to learn richer, more generalizable video representations. Notably, FLF2V and FLC2V serve as temporal regularizers, counteracting dynamics degradation seen with other auxiliary tasks alone. Visual illustrations are presented in the Appendix.

Finally, removing depth conditions consistently degrades performance across all metrics, including Dynamics ($-10\%$), Imaging Quality ($-4\%$), and Scene Consistency ($-12\%$). This demonstrates that geometric cues from the 3D depth map serve as a powerful complement to multi-task learning, significantly enhancing motion dynamics, perceptual quality, and scene-level coherence.

## 5 CONCLUSION

In this work, we introduced MfM (Many-for-Many), a unified video foundation model capable of handling diverse visual generation and manipulation tasks through a single parameter-efficient architecture. Specifically, we designed a lightweight adapter to effectively unify various 2D and 3D conditions into a uniform representational space, enabling seamless integration into our video generation pipeline. By employing progressive joint image-video learning and multi-task training strategies, we not only enabled multiple visual generation and manipulation capabilities within a single model but also transferred the knowledge from other image tasks to video generation. This knowledge sharing significantly reduced the required amount of costly text-to-video training data and enhanced the fundamental video generation capabilities. As validated in our experiments, MfM achieved competitive or superior performance compared to specialized models and even commercial systems while using much fewer training data and model parameters.

**Limitations**. Despite the demonstrated effectiveness, we acknowledge certain limitations of our proposed MfM. Currently, MfM processes 1D conditions (text) and 2D/3D conditions (masks, pixels, depth) separately before implicitly fusing them through self-attention in DiT blocks. In future work, we will explore the use of vision-language models rather than text-only encoders to perform explicit multimodal fusion earlier in the pipeline, which could enhance performance on tasks requiring comprehensive understanding of complex input conditions.

## 6 ETHICS STATEMENT

This work does not involve any human subjects or sensitive personal data. The usage of all datasets strictly complies with their respective licenses.

Our methods are intended solely for academic and scientific purposes. We do not foresee direct harmful applications, but acknowledge that misuse could occur if applied without proper safeguards. We encourage responsible use of the research outcomes, with attention to fairness, transparency, and legal compliance.

## 7 REPRODUCIBILITY STATEMENT

We have taken several measures to ensure the reproducibility of our work. All details of the proposed model, preprocessing steps of datasets and algorithms with full hyperparameter settings and training procedures provided are described in the main text.

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

## A  APPENDIX

In this appendix, we provide visual demonstrations and the following supporting materials to the main paper:

- Declaration of LLM Assistance;
- The hyper-parameters and inference latency of 2B&8B MfM (referring to Sec. 3 in the main paper);
- The details of our data augmentation strategy and MfM-Benchmark construction (referring to Sec. 4.1 in the main paper);
- Visual results of T2V generation on VBench (referring to Sec. 4.2 in the main paper);
- Visual results of I2V generation on VBench (referring to Sec. 4.2 in the main paper);
- User study of I2V generation;
- Visual results of multi-task generation on MfM-Benchmark (referring to Sec. 4.3 in the main paper);
- Visual results of T2V generation with or without multi-task training (referring to Sec. 4.4 in the main paper);
- Failure cases of MfM;
- Details about training data;
- Ablation study on sampling probability;
- Evaluation results on MovieGen Benchmarks;
- Ablation study on task interactions between different tasks;
- Quantitative Gains of Q-K Normalization and 3D RoPE;
- Ablation study on model adaptation to new tasks.

For better viewing experience, we uploaded the video demos to a dedicated anonymous website https://anonymous.4open.science/w/MfMPage-2602/, where the videos can be played directly in the browser. Note that, due to the significant number of high-quality video files included in our demonstrations, initial page loading may require several minutes to complete. We appreciate your patience during this process, as the complete visual experience is essential to understand the capabilities and performance of our approach.

### A.1  DECLARATION OF LLM ASSISTANCE

We use ChatGPT-5 to assist with the refinement of this manuscript. After drafting the full text, we provided selected passages to the models for suggestions on grammar, clarity, and conciseness. All revisions were reviewed and finalized by the authors to ensure accuracy and appropriateness.

### A.2  THE HYPER-PARAMETERS AND INFERENCE LATENCY OF 2B&8B MFM

In Table 7, we present the detailed hyper-parameter settings of our two MfM variants. The larger model has 8 billion parameters with 40 layers, 48 attention heads, and a hidden dimension of 3,072, whereas the smaller model has 2 billion parameters with 28 layers, 28 attention heads, and a hidden dimension of 1,792.

Table 7: Hyper-parameters of our 2B and 8B model variants.

| Model Size | Layers | Attention Heads | Head Dim | FFN Dim | Cross-Attn Dim |
|------------|--------|-----------------|----------|---------|----------------|
| 2B | 28 | 28 | 64 | 7168 | (1792, 2048) |
| 8B | 40 | 48 | 64 | 12288 | (3072, 2048) |

Table 8: Model Inference Time of 2B&8B MfM

| Model | Resolution | Steps | Time |
|-------|------------|-------|------|
| 2B | [97,128,224] | 30 | 5.80s |
| 2B | [97,360,640] | 30 | 16.12s |
| 2B | [97,720,1280] | 30 | 2:05 |
| 8B | [97,128,224] | 30 | 9.03s |
| 8B | [97,360,640] | 30 | 32.90s |
| 8B | [97,720,1280] | 30 | 4:19 |

In Table 8, we report the inference latency of our model under different resolutions. For inference, we adopt sequence parallelization and Teacache (Liu et al., 2024) to improve efficiency. Notably, due to the unified adapter interface and the simple additive integration of adapter outputs into the latent features, the inference cost remains nearly constant across tasks.

## A.3 THE DETAILS OF MFM-BENCHMARK CONSTRUCTION

For multi-task data augmentation strategy, we applied the following enhancement pipeline:

1. Text-to-Video (T2V): We used the original captions as conditioning input.

2. Image-to-Video (I2V): We used the first frame and caption as conditioning input.

3. Video Extension (VEXT): We extracted the first 8 frames as conditioning input to generate the remaining frames.

4. Video Inpainting (VINP): We applied random masks to interior regions covering 1/9 to 1/4 of the total pixels.

5. Video Outpainting (VOUTP): We generated boundary masks covering 1/8 to 1/4 of the total width/height.

6. Video Colorization (VCOLOR): We converted the ground-truth videos to grayscale.

7. First-Last-Frame-to-Video (FLF2V): We used the first and last frames as conditioning input to generate the intermediate 95 frames.

8. First-Last-Clip-to-Video (FLC2V): We used the first 8 frames and last 8 frames as conditioning input.

9. Video Super-Resolution (VSR): We applied random downsampling factors between $2\times$ and $6\times$ and used the downsampled videos as conditioning input.

10. Video Editing (VEDIT): We used the original videos as conditioning input, replacing the original captions with style instruction prompts (*e.g.*, "change the video to oil painting style").

11. Text-to-Image (T2I): We used the first frame at the ground-truth.

12. Image Super-Resolution (SISR): We used the first frames at the ground truth an downscaled them with downsampling factors between $2\times$ and $6\times$.

13. Image Inpainting (IINP): We sampled the first frames and randomly masked them like VINP.

14. Image Outpainting (IOUTP): we sampled the first frames and randomly masked them like VOUTP.

15. Image Coloraization (ICOLOR): We sampled the first frames and converted them to grayscale.

16. Image Editing (IEdit): We sampled the first frames and replaced the original captions with instruction prompts such as VEDIT.

For MfM-benchmark, first, we collected 1500 videos of 1280×720 of resolution and their accompanying captions from Pexels (Pexels, 2025), selecting only those containing more than 97 frames. We then applied a two-stage quality filtering process: (1) removing blurry videos by calculating the CV2.Laplacian (Bradski, 2000) score for each frame and excluding those below a threshold of 200, and (2) evaluating motion dynamics using RAFT (Teed & Deng, 2020) and retaining only videos with motion scores exceeding 5. This filtering resulted in our final dataset of 480 high-quality videos, which serve as ground-truth for reference-based metrics. We standardized each video to 97 frames and divided them into 16 segments for consistent processing. Final, we applied the above enhancement pipeline to prepare the condition.

Visual illustrations of these tasks are shown in Figure 5, Figure 14, Figure 15, Figure 16, Figure 17. Video demonstrations are also available at https://anonymous.4open.science/w/MfMPage-2602/.

## A.4 VISUAL RESULTS OF T2V GENERATION ON VBENCH

Our comprehensive qualitative analysis spans four diverse text-to-video generation scenarios—coastal beach oil painting with waves, person walking in snowstorm, koala playing piano in forest, and bicycle slowing down—as illustrated in Figures 6, 7, 8 and 9. We compare MfM with Wan2.1 (Team, 2025), Hunyuan (Kong et al., 2024), Sora (OpenAI, 2024), Gen-3 (RunwayML, 2023), PikaLabs (Labs, 2023), LTX-Video (HaCohen et al., 2024), CogVideoX1.5 (Yang et al., 2024c), and EasyAnimate (Xu et al., 2024). From these figures and the demo videos in our provided anonymous website, we can have the following observations.

Wanx2.1 exhibits prompt comprehension failures across multiple dimensions. For example, it fails to capture motion elements—generating a moving bicycle in Figure 9 that shows no deceleration, and even producing backwards walking motion in Figure 7, contradicting natural human movement. Hunyuan produces near-identical frames in the beach scene (Figure 6), where waves show negligible movement. Additionally, Hunyuan demonstrates limited stylistic interpretation capability, completely missing the oil painting aesthetic in Figure 6. Sora produces significant contextual mismatches in several scenarios. Notably, it generates an urban nighttime scene instead of a snowstorm in Figure 7. While Sora delivers reasonable visual quality, it frequently produces minimal frame-to-frame progression, which is particularly evident in the bicycle sequence, where speed reduction is barely perceptible. Gen-3 generally provides good visual quality but struggles with specific prompt elements. It fails to accurately render koala coloration in Figure 8, producing an unnatural scenario where the koala is in the piano. In Figure 9, it shows a riderless bike with non-diminishing dust effects that physically contradict the slowing action specified in the prompt.

PikaLabs demonstrates framing issues across multiple scenarios. In Figure 7, the human subject appears too small to effectively convey walking motion. This problem is even more pronounced in Figure 9, where an inappropriately wide urban composition makes the bicycle barely visible. LTX-Video exhibits the most severe quality limitations, consistently delivering washed-out, minimalist renderings across all scenarios. Most problematically, LTX-Video demonstrates dramatic mid-sequence discontinuities in Figure 8, completely changing the scene halfway through. CogVideoX generates video with small motion changes and cannot adapt to the style prompting (Figure 6). Easyaimate completely misidentifies the requested animal in Figure 8, rendering a panda instead of a koala. In Figure 9, it shows an inappropriate close-up framing of a stationary bicycle wheel, making the slowing action impossible to perceive. In contrast, MfM demonstrates superior results across all scenarios, achieving an ideal balance of prompt fidelity, motion physics, and visual quality.

## A.5 VISUAL RESULTS OF I2V GENERATION ON VBENCH

Our qualitative analysis spans four diverse cases—swimming turtle, dog carrying a soccer ball, fishing boat navigation, and galloping horses—as illustrated in Figures 10, 11, 12 and 13. These scenarios were selected to evaluate model performance across a spectrum of challenges, including animal locomotion, object interaction, environmental dynamics, and atmospheric conditions. We compare our MfM with Wanx-I2V (Team, 2025), Hunyuan-I2V (Kong et al., 2024), Magi-1 (Sand-AI, 2025), Step-Video (Ma et al., 2025), DynamicCrafter (Xing et al., 2023), VideoCrafter-I2V (Chen et al., 2024a), I2VGen-XL (Zhang et al., 2023b), CogvideoX-I2V (Yang et al., 2024c), ConsistI2V (Ren et al., 2024). From these figures and the demo videos in our provided anonymous website, we can have the following observations.

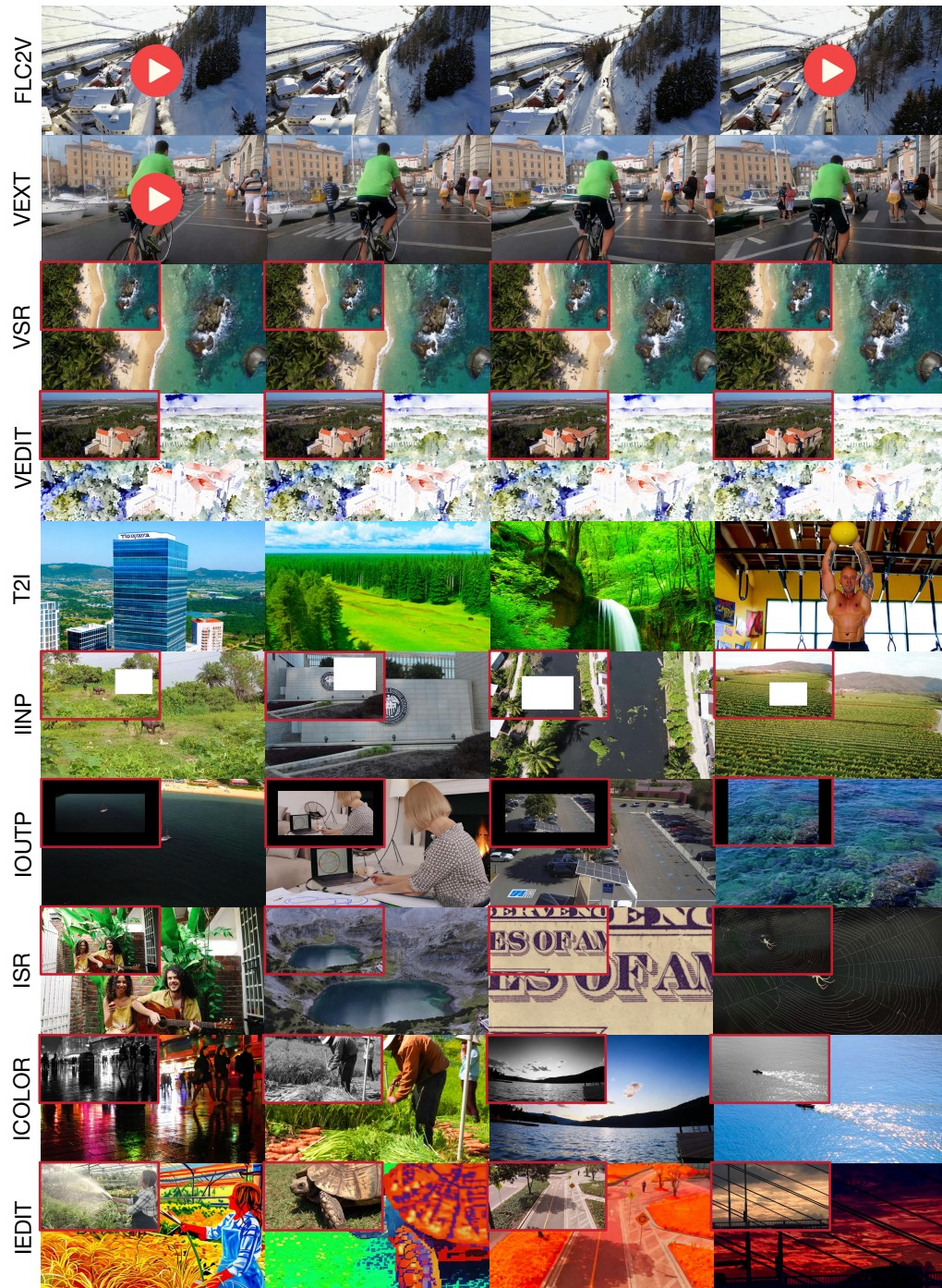

Figure 5: Visual illustrations for different tasks supported by our MfM.

Hunyuan-I2V demonstrates minimal temporal progression across all scenarios, producing sequences with negligible motion variation. This is particularly evident in the turtle (Figure 10) and fishing boat (Figure 12) examples. Furthermore, Hunyuan-I2V introduces anatomical inconsistencies in the horse sequence, rendering equine subjects with only three legs in later frames—a critical biological implausibility. Wanx-I2V can produce reasonable animal movement, but sometimes fail to capture essential action descriptors. For example, it fails to generate the "navigating" movement explicitly specified in the boat prompt (Figure 12). StepVideo-I2V suffers from visual artifacts across multiple dimensions, including anatomical anomalies (abnormal turtle fin articulation in Figure 10), subject identity inconsistencies (altered dog appearance in Figure 11), and most strikingly, fundamental scene

Table 9: User votes on 10 image-to-video generation outputs

| Method | MfM | Wanx I2V | Hunyuan I2V | StepVideo I2V | Magi-1 | Cogvideo I2V | I2VGenXL | DynCrafter | VidCrafter | ConsistI2V |
|---|---|---|---|---|---|---|---|---|---|---|
| Top-1 Rates | 64.29% | 20.00% | 1.43% | 4.29% | 0.00% | 5.71% | 4.29% | 0.00% | 0.00% | 0.00% |
| Top-3 Rates | 85.71% | 71.43% | 40.00% | 41.43% | 12.86% | 30.00% | 12.86% | 0.00% | 4.29% | 1.43% |

misinterpretation in the horse sequence. CogVideo-I2V demonstrates object consistency failures, including problematic size variations in the turtle sequence and unstable object interactions in the dog example. DynamiCrafter exhibits even more pronounced temporal instability, with objects and environmental elements changing unnaturally between consecutive frames—most evident in the inconsistent appearance of soccer ball and geometric distortions of the dog subject in Figure 11. VidCrafter and ConsistI2V both struggle with maintaining prompt fidelity, frequently altering the fundamental identity characteristics in the conditioning image. This prompt deviation is particularly pronounced in the dog sequence (Figure 11), where breed characteristics, coat patterns, and contextual elements shift significantly from the reference image. In contrast, our MfM and Magi-1 achieve both robust identity preservation and convincing motion dynamics in all test cases.

## A.6 USER STUDY OF I2V GENERATION

To comprehensively evaluate MfM's effectiveness in generative tasks, we conduct a user study specifically focused on image-to-video (I2V) generation, comparing against nine I2V generation methods whose models are publicly available: Wanx-I2V (Team, 2025), Hunyuan-I2V (Kong et al., 2024), StepVideo-I2V (Ma et al., 2025), Magi-1 (Sand-AI, 2025), Cogvideo-I2V (Yang et al., 2024c), I2VGenXL (Zhang et al., 2023b), DynamiCrafter (Xing et al., 2023), VideoCrafter (Chen et al., 2024a), and ConsistI2V (Ren et al., 2024). The user study comprised 10 test cases encompassing various content categories, including animal motion, human activities, scenic close-ups, and vehicular movement. We invited 10 participants and asked them to rank the top three generated videos for each case based on visual quality and semantic consistency. Table 9 presents the average Top-1 and Top-3 rates for all methods. The results clearly show that MfM outperforms all competitors, achieving a 64.29% Top-1 rate and an 85.71% Top-3 rate. Wanx-I2V ranks second with 20.00% Top-1 and 71.43% Top-3 rates, respectively. Hunyuan-I2V and StepVideo-I2V demonstrate moderate performance with Top-3 rates of approximately 40%, despite Top-1 rates below 5%. Notably, Magi-1, DynamiCrafter, VideoCrafter, and ConsistI2V fail to secure any Top-1 selections and exhibit minimal presence in Top-3 rankings. These results reveal MfM's superior capability in generating high-quality image-to-video content that consistently meets human evaluation criteria.

## A.7 VISUAL RESULTS OF MULTI-TASK GENERATION

The visual results of four representative video tasks, including VINP, VOUTP, VCOLOR and FLF2V, are illustrated in Figure 14, Figure 15, Figure 16, Figure 17, respectively.

For VINP, we compare our MfM with VACE (Jiang et al., 2025) and ProPainter (Zhou et al., 2023) across three diverse scenarios in Figure 14. We see that MfM demonstrates superior performance in maintaining visual fidelity and temporal consistency. Specifically, both MfM and VACE produce reasonably coherent results in the first and third cases, where they successfully reconstruct the masked region with detail preservation and natural integration with the surrounding environment. However, in the second case, VACE shows inconsistencies in intensity distribution and color. ProPainter exhibits severe blurring and artifacts in the inpainted region, failing to properly reconstruct the subject and completely losing the details.

For VOUTP, we compare MfM with VACE (Jiang et al., 2025), Follow-Your-Canvas (FYC) (Chen et al., 2024b), and M3DDM (Fan et al., 2023). The comparisons on three diverse scenarios are illustrated in Figure 15. MfM demonstrates exceptional consistency and contextual understanding across all test cases. VACE shows moderate capabilities but with noticeable limitations. While it produces acceptable wave continuation in the ocean scene, it generates noticeable brightness mismatches between the original and generated regions in the second case. FYC suffers from the brightness mismatches in the second case; what's more, it fails to complete the leg of the person in the first frame of the first case. M3DDM exhibits significant limitations in this task. It generates blurred outputs and visually jarring discontinuities around the generated areas.

For VCOLOR, we compare MfM with Colormnet (Yang et al., 2024b) and TCVC(Zhang et al., 2023c) in Figure 16. We see that MfM demonstrates excellent contextual understanding performance and maintains superior temporal color stability between adjacent frames. Compared with other baselines, it achieves more complete colorization coverage without introducing grayscale artifacts while preserving realistic lighting conditions. Colormnet shows reasonable performance on these cases but suffers from saturation issues in the last two cases. TCVC exhibits substantial limitations across all test scenarios, with large portions remaining in grayscale and the overall color tone appearing excessively dull and lifeless.

For FLF2V, we compare MfM with Wanx-FLF2V (Team, 2025) and Hunyuan (with keyframe LoRA) (Kong et al., 2024). The comparisons on three scenarios are illustrated in Figure 17. We see that both MfM and Wanx-FLF2V deliver natural motion interpolation between the first frame and the last frame without jarring transitions, as shown in the first and third cases. But Wanx-FLF2V performs abnormally in the second case, where the video frames are unexpectedly compressed vertically at the end, altering the aspect ratio. Hunyuan exhibits severe limitations in the first and third cases. It produces intermediate frames with a different viewpoint and a noticeably darkened color tone, resulting in jarring visual transitions.

## A.8 VISUAL RESULTS OF ABLATION STUDY

The ablation study results on four scenarios are illustrated in Figure 18, which provides visual evidence to support that multi-task training significantly improves the temporal dynamics of the generated videos. We can see that T2V w/ MfM demonstrates cinematographic qualities, including smooth and flexible camera movements, as well as vivid and evolving patterns. For instance, in the first case, the varying angle of the video effectively captures the dynamic essence of 'gain speed'; in the second case, dynamic color transitions and the natural progression of pyrotechnic effects are illustrated; in the third case, the train is portrayed with appropriate motion blur; and in the fourth, the celestial progression is dramatically captured, with the sun emerging and intensifying across the horizon, accompanied by corresponding atmospheric lighting changes. In contrast, T2V w/o MfM exhibits minimal camera movement, with limited perspective variation and an almost static side view throughout the sequence. Furthermore, in the last case, T2V w/o MfM produces nearly identical frames of a static sun, with little temporal progression.

## A.9 FAILURE CASES

While MfM demonstrates strong performance across T2V and I2V generation tasks, it also occasionally produces failure cases, as illustrated in Figure 19. First, in complex interaction scenarios, MfM may produce physically implausible object relationships. For instance, in the basketball dunking sequence (first row), the ball incorrectly traverses the net rather than entering the basket properly. Similarly, in the burger eating sequence (second row), the burger wrapper abruptly merges into the burger in the intermediate frames. Second, MfM also exhibits limitations in generating videos that contain words; this is particularly evident in the cyberpunk cityscape (third row) and the animated panda scene (fourth row). Finally, for sequences involving rapid motion, we observe temporal artifacts manifesting as duplicated or misplaced features. This is exemplified in the cat playing sequence (fifth row), where an anomalous second tail-like object appears near the cat's head in intermediate frames. Meanwhile, in the sword fighting sequence (last row), the character on the right undergoes noticeable variations and distortions in the intermediate frames. Future work will be conducted to further improve the performance of MfM on these scenarios.

## A.10 DETAILS ABOUT TRAINING DATA

Regarding our training data, over 70% of them are collected from publicly available sources, including Panda70M (Chen et al., 2024c), Koala36M (Wang et al., 2025), InternVid Wang et al. (2023), OpenVid (Nan et al., 2024), and WebVid (Bain et al., 2021), complemented by a small portion of proprietary data. For images, the primary source is LAION-5B (Schuhmann et al., 2022).

To ensure data quality, we adopt a multi-stage filtering pipeline:

- **Video Segmentation:** We first apply PySceneDetect for coarse scene boundaries. Then, we extract frame-level features using DINOv2 (Oquab et al., 2023), compute inter-frame

similarity, and further split clips at low-similarity points. Videos shorter than 2 seconds are removed.

- **Video Quality Filtering:** Each segmented clip is evaluated along several dimensions: 1) basic metadata (FPS, resolution, bitrate) extracted directly from video; 2) average aesthetic score using a pretrained aesthetic model; 3) overlay text ratio via a pretrained OCR model; 4) watermark detection through a dedicated watermark model; 5)for motion quality, we compute optical flow using RAFT (Teed & Deng, 2020), then filter out clips with insufficient motion.

- **Semantic Content Filtering:** To identify and remove potential low-quality or undesirable content, we employ a fine-tuned VideoLLaMA3 (Zhang et al., 2025) model to detect unsafe content, low-light or blurry scenes, overexposed frames, black borders, abrupt perspective shifts, and static-image animations.

- **Video Captioning:** For caption generation, we use Tarsier2 (Yuan et al., 2025), prompting it to produce two complementary captions: a short global summary and a long, detailed description. These two captions are merged to form the final caption for each clip.

## A.11 ABLATION STUDY ON SAMPLING PROBABILITY

We chose to assign 3× higher sampling probability to the basic generation tasks for two main reasons. First, these three tasks (T2V, I2V, and T2I) represent the core generation capabilities most commonly required in practical scenarios. Unlike editing tasks, which provide strong and explicit conditioning signals, these basic generation tasks rely on weaker supervision and are therefore significantly harder to optimize. Allocating additional sampling probability ensures that the backbone generative ability is sufficiently strengthened during pretraining.

Second, we conducted an ablation study to investigate how different sampling ratios affect model performance. Specifically, we compared four settings: 1) Equal sampling probability across all tasks 2) 2× sampling probability for the three basic tasks 3) 3× sampling probability for the three basic tasks 4) 4× sampling probability for the three basic tasks.

The evaluation results on VBench-T2V are shown in the Table 10. Among all configurations, the 3× sampling strategy consistently achieves the strongest overall performance across most metrics, demonstrating that an appropriately biased multi-task sampling schedule can effectively enhance generative capability without increasing the training budget.

In contrast, sampling ratios that allocate insufficient training budget to the basic generation tasks (*e.g.*, 1× or 2×) lead to under-optimized T2V performance. In these settings, the T2V task does not receive enough updates to fully benefit from the complementary supervision provided by other tasks.

Conversely, overemphasizing the basic tasks (*e.g.*, 4×) weakens the regularization effect brought by the editing tasks. This reduces multi-task synergy and results in performance degradation across several metrics.

As an extreme case, assigning zero probability to all other tasks degenerates the training back to a pure T2V paradigm, which—as demonstrated in the Table 6 of the main paper—performs worse than our mixed MfM training framework. This further validates that the improvements are not solely due to the basic tasks themselves, but arise from the interaction among diverse tasks under a well-balanced sampling strategy.

Table 10: Quantitative comparison of T2V generation performance with different sampling probability on the VBench-T2V benchmark. For each dimension, the best result is in bold, the second best result is underscored.

| Model | Motion | Dynamic | Aesth. | Img. | Object | Mul.Obj. | Spatial | Scene | Appear. | Temp. | Consist. |
|---|---|---|---|---|---|---|---|---|---|---|---|
| 1× | 0.9865 | 0.6528 | 0.5822 | 0.5493 | 0.8726 | **0.6395** | **0.6026** | 0.4789 | 0.2254 | 0.2299 | **0.2507** |
| 2× | 0.9664 | 0.8750 | 0.5520 | 0.5420 | 0.7642 | 0.3361 | 0.4578 | **0.4942** | 0.2285 | 0.2272 | 0.2495 |
| 3× | **0.9922** | **0.8889** | **0.5911** | **0.5853** | **0.8861** | 0.5160 | 0.5851 | 0.4869 | **0.2289** | 0.2341 | 0.2483 |
| 4× | 0.9810 | 0.6667 | 0.5834 | 0.5791 | 0.8441 | 0.4177 | 0.4977 | 0.4680 | 0.2277 | **0.2393** | 0.2470 |

## A.12 EVALUATION RESULTS ON MOVIEGEN BENCHMARKS

Besides VBench, we also adopt another widely used benchmark — the MovieGen Benchmark (Polyak et al., 2025) released by Meta (hereafter referred to as the MovieGen Benchmark) — for further evaluation. This benchmark provides broader coverage across key evaluation dimensions and includes diverse motion categories (*i.e.*, high, medium, and low motion prompts). It has also been adopted by recent state-of-the-art works such as Veo 3 (Google, 2025) and Goku (Chen et al., 2025). Since the MovieGen Benchmark does not provide official evaluation metrics, we employ the same set of video quality evaluation metrics used in VBench to measure the performance of different models on several key metrics. The results of open-sourced models are presented in Table 11.

Table 11: Quantitative comparison of T2V generation performance on the MovieGen benchmark.

| Model | Aesthetic. | Imaging | Dynamic | Temporal | Consistency | Avg. Rank. |
|---|---|---|---|---|---|---|
| MfM (8B) | 0.6136 | 0.6887 | **0.7188** | 0.2615 | 0.2615 | **2.2** |
| Wanx (14B) | **0.6428** | **0.7265** | 0.2969 | 0.2474 | 0.2474 | 3.4 |
| Hunyuan (13B) | 0.6044 | 0.6483 | 0.5469 | 0.2651 | **0.2651** | 2.4 |
| Opensora (11B) | 0.6311 | 0.6180 | 0.5156 | **0.2688** | 0.2588 | 2.6 |
| Cogvideo (5B) | 0.5634 | 0.6104 | 0.4844 | 0.2452 | 0.2452 | 4.8 |
| EasyAnimate (12B) | 0.5439 | 0.5697 | 0.4322 | 0.1375 | 0.1375 | 6.2 |
| LTX-video (0.98B) | 0.5087 | 0.5603 | 0.3438 | 0.2063 | 0.2063 | 6.8 |

The results on the MovieGen benchmark further demonstrate the advantages of our MfM. In particular, MfM (8B) achieves the best performance on video dynamics and the second-best results on image quality and overall consistency, closely matching or even surpassing those much larger models such as Wanx (14B), Hunyuan (13B), and OpenSora (11B). Note that our **MfM model achieves this performance by training on only 160M images and 120M video clips**, far less than those models like Wanx and Hunyuan, which are trained on billion-scale datasets.

Regarding long-duration video generation, our model is primarily trained on clips of 97 frames. Extending the temporal window significantly increases the computational and memory cost during training, and thus long-duration generation is currently beyond the intended scope of this work. A promising direction is to adapt our MfM framework to a streaming or chunk-wise generation pipeline, which would enable arbitrarily long videos. We consider this as an important extension in future work.

## A.13 ABLATION STUDY ON TASK INTERACTIONS BETWEEN DIFFERENT TASKS

Given the large number of possible task combinations, it is infeasible to conduct an exhaustive ablation over all of them. Therefore, we select a subset of representative tasks (T2V, VINP, FLF2V, VColor) to study task interactions. In particular, starting from a T2V-only checkpoint, we evaluate mixing strategies including: T2V only, T2V + any single task, T2V + any two tasks, and T2V + all selected tasks. The evaluation results on VBench-T2V are summarized in Table 12:

Table 12: Quantitative comparison of different task interactions. For each dimension, the best result is in bold, the second best result is underscored.

| Model | Motion | Dynamic | Aesth. | Img. | Object | Mul.Obj. | Spatial | Scene | Appear. | Temp. | Consist. |
|---|---|---|---|---|---|---|---|---|---|---|---|
| T2V | 0.9915 | 0.6556 | 0.5931 | 0.5448 | 0.8869 | 0.5457 | 0.6112 | 0.5065 | 0.2214 | 0.2379 | 0.2520 |
| T2V+VCOLOR | 0.9892 | 0.6111 | 0.5881 | 0.5484 | 0.8339 | 0.5122 | 0.4326 | 0.5000 | 0.2317 | 0.2406 | 0.2519 |
| T2V+VINP | 0.9887 | 0.6944 | 0.5770 | 0.5402 | 0.8703 | 0.5655 | 0.5714 | 0.4833 | 0.2333 | 0.2288 | 0.2474 |
| T2V+FLF2V | 0.9911 | 0.7778 | 0.5930 | 0.5437 | **0.8932** | **0.6006** | 0.5098 | 0.4935 | 0.2295 | 0.2415 | 0.2464 |
| T2V+FLF2V+VCOLOR | 0.9846 | 0.7639 | 0.5743 | 0.5725 | 0.8623 | 0.5358 | **0.6515** | 0.5291 | 0.2298 | 0.2419 | 0.2551 |
| T2V+VINP+FLF2V | **0.9925** | 0.8472 | **0.5947** | 0.5808 | 0.9090 | 0.5983 | 0.5261 | **0.5356** | 0.2291 | **0.2421** | 0.2551 |
| T2V+VINP+VCOLOR | 0.9870 | 0.7778 | 0.5816 | 0.5785 | 0.8576 | 0.5816 | 0.5150 | 0.5007 | **0.2347** | 0.2333 | 0.2445 |
| T2V+FLF2V+VCOLOR+VINP | 0.9922 | **0.8750** | 0.5911 | **0.5853** | 0.8861 | 0.5760 | 0.5851 | 0.4969 | 0.2289 | 0.2372 | **0.2553** |

As shown in the table, in most cases, when both tasks improve performance on certain metrics, incorporating them into the mixed training pipeline also brings benefits (*e.g.*, dynamic, appearance). However, when one task improves performance while another negatively affects it, the final mixed

results vary across metrics. For example, for object and multiple object metrics, integrating FLF2V leads to significant gains. When further combining it with VCOLOR, the mixed model still improves upon the T2V baseline, but the magnitude of improvement narrows because VCOLOR has a mild negative effect on single-object and multi-object generation accuracy. Conversely, mixing VINP and VCOLOR with T2V can degrade performance on temporal style, though VCOLOR alone improves general generation quality.

Interestingly, we also observe cases where two individually harmful tasks produce unexpected improvements when combined. For instance, VINP and FLF2V each weaken some metrics, but together, they significantly improve scene and overall consistency. We attribute this to the complementary regularization effects they impose on holistic video understanding: although VINP primarily focuses on spatial completion and FLF2V on temporal completion, each task alone may bias optimization in an unbalanced direction, whereas their combination better constrains the model and leads to improved global metrics like overall consistency.

Finally, when all tasks are included in training, the model achieves substantial gains on most metrics (*e.g.*, motion, dynamic, image quality, appearance, overall consistency). Some metrics drop slightly compared to T2V-only training due to the absence of regularization from tasks that specifically benefit them. For example, as shown in Table 6 of the main paper, metrics such as aesthetics and scene benefit greatly from incorporating video-extension and first-last-clip-to-video tasks.

However, it is difficult to accurately evaluate the effect of each individual task and all possible combinations, given the enormous combinatorial space and the complex interactions among tasks. Therefore, throughout this work, we focus on assessing the overall performance of the model under mixed-task training—examining whether the model can simultaneously learn 10+ editing capabilities while leveraging their interactions to improve video generation ability.

### A.14 QUANTITATIVE GAINS OF Q-K NORMALIZATION AND 3D RoPE

To better clarify the contribution of Q–K Normalization and 3D Rotary Position Embedding (3D RoPE), we performed an ablation study by finetuning a 2B-parameter checkpoint for an additional 10K training steps under three settings: (1) full model, (2) removing Q–K Norm, and (3) removing 3D RoPE. We report the results on the VBench-T2V benchmark in Table 13.

Table 13: Quantitative gain of Q-K normalization and 3D Rotary Position Embedding.

| Model | Motion | Dynamic | Aesth. | Img. | Object | Mul.Obj. | Spatial | Scene | Appear. | Temp. | Consist. |
|---|---|---|---|---|---|---|---|---|---|---|---|
| MfM-baseline | 0.9922 | 0.8889 | 0.5911 | 0.5853 | 0.8861 | 0.5160 | 0.5851 | 0.4869 | 0.2289 | 0.2341 | 0.2483 |
| MfM (w/o Q-K Norm) | NA | NA | NA | NA | NA | NA | NA | NA | NA | NA | NA |
| MfM (w/o RoPE) | 0.9659 | 0.8472 | 0.3149 | 0.3628 | 0.0973 | 0.0000 | 0.0089 | 0.0022 | 0.2200 | 0.0423 | 0.0584 |

In our experiments, the model consistently collapses after 3K steps once Q–K Normalization is removed. This collapse manifests as exploding attention activations and rapidly diverging losses, preventing further training. This confirms that Q–K Norm is critical for stabilizing large-scale DiT-based video generation models, especially under our multi-task many-for-many training regime where diverse conditioning signals create additional gradient variance.

Unlike Q–K Norm, removing 3D RoPE does not cause training divergence; however, it leads to substantial degradation across all VBench metrics. The drop is particularly severe for spatial–semantic metrics such as Object, Multiple Objects, Spatial, and Scene. Without 3D RoPE, the model frequently fails to place objects in correct spatial locations or maintain consistent geometry throughout the video, resulting in near-zero performance on these categories. This demonstrates that 3D RoPE is crucial for modeling the joint spatial–temporal structure of video tokens.

### A.15 ABLATION STUDY ON MODEL ADAPTATION TO NEW TASKS.

To evaluate the models adaptation capacity after multi-tasks training, we performed the following experiment: starting from the original T2V model, we conducted 10K-step T2V-only training (Model 1) and multi-task training (Model 2, excluding VINP tasks). After training, we fine-tuned both models on the new VINP task and compared their adaptation performance. The results, shown in the

A beautiful coastal beach in spring, waves lapping on sand, oil painting

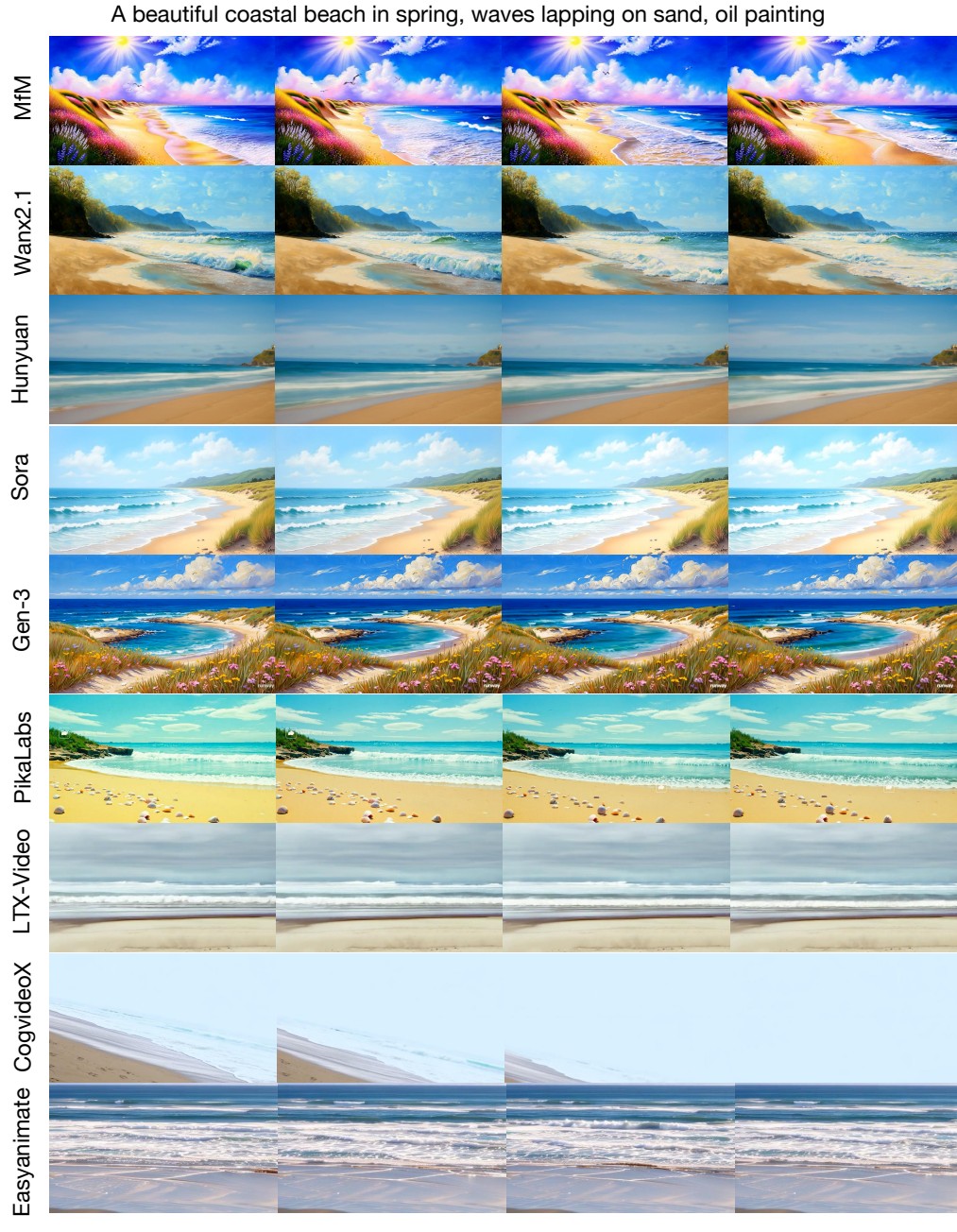

Figure 6: T2V generated videos with prompt "a beautiful coastal in spring, waved lapping on sand, oil painting".

Table 14, demonstrate that the multi-task model adapts more effectively to the new task, highlighting the benefits of our unified multi-task pretraining approach.

Table 14: Performance comparison on video inpainting task.

| Method | Reference-based Metrics | | | | No-reference Perceptual Metrics | | | | |
|--------|------|------|------|--------|--------|------|--------|----------|-------|
| | FVD↓ | PSNR↑ | SSIM↑ | LPIPS↓ | Aesth.↑ | Img.↑ | Motion↑ | Consist.↑ | Temp.↑ |
| Model 1 | 758.894 | 28.740 | 0.699 | 0.296 | 0.5284 | 0.7201 | 0.9940 | 0.2258 | 0.9894 |
| Model 2 | 740.026 | 29.113 | 0.709 | 0.286 | 0.5339 | 0.7347 | 0.9939 | 0.2281 | 0.9890 |

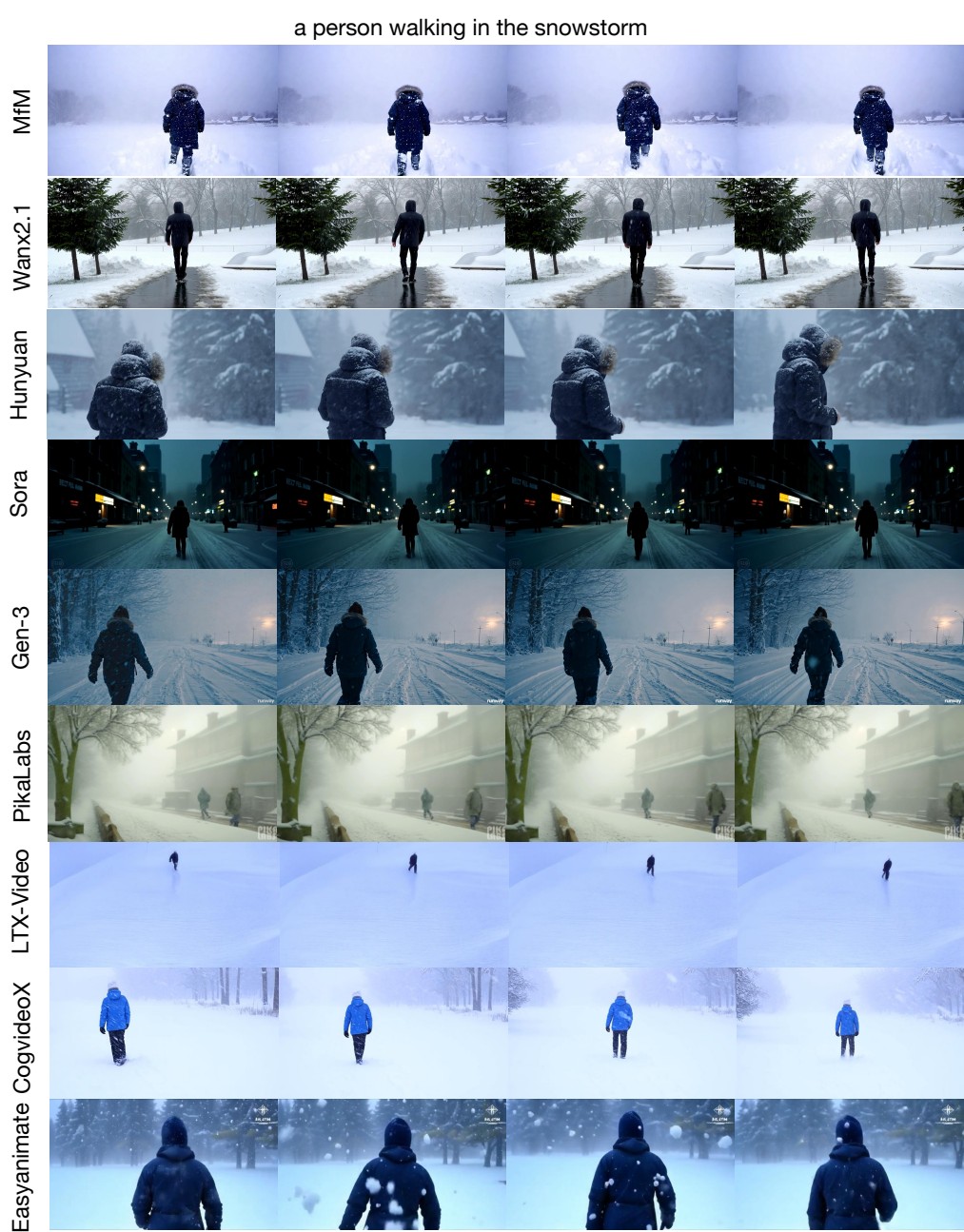

Figure 7: T2V generated videos with prompt "a person walking in the snowstorm".

A koala bear playing piano in the forest

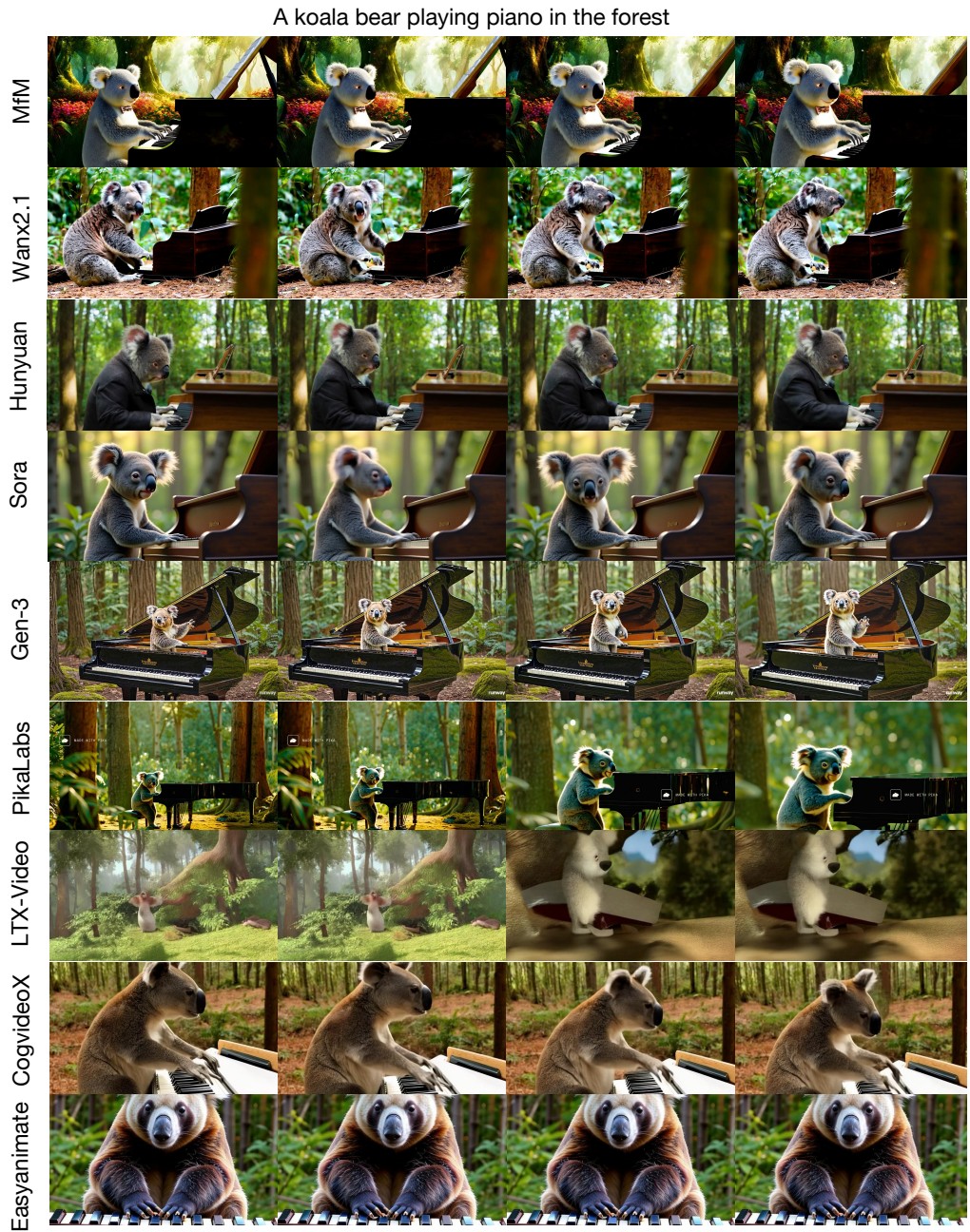

Figure 8: T2V generated videos with prompt "a koala bear playing piano in the forest".

A bicycle slowing down to stop

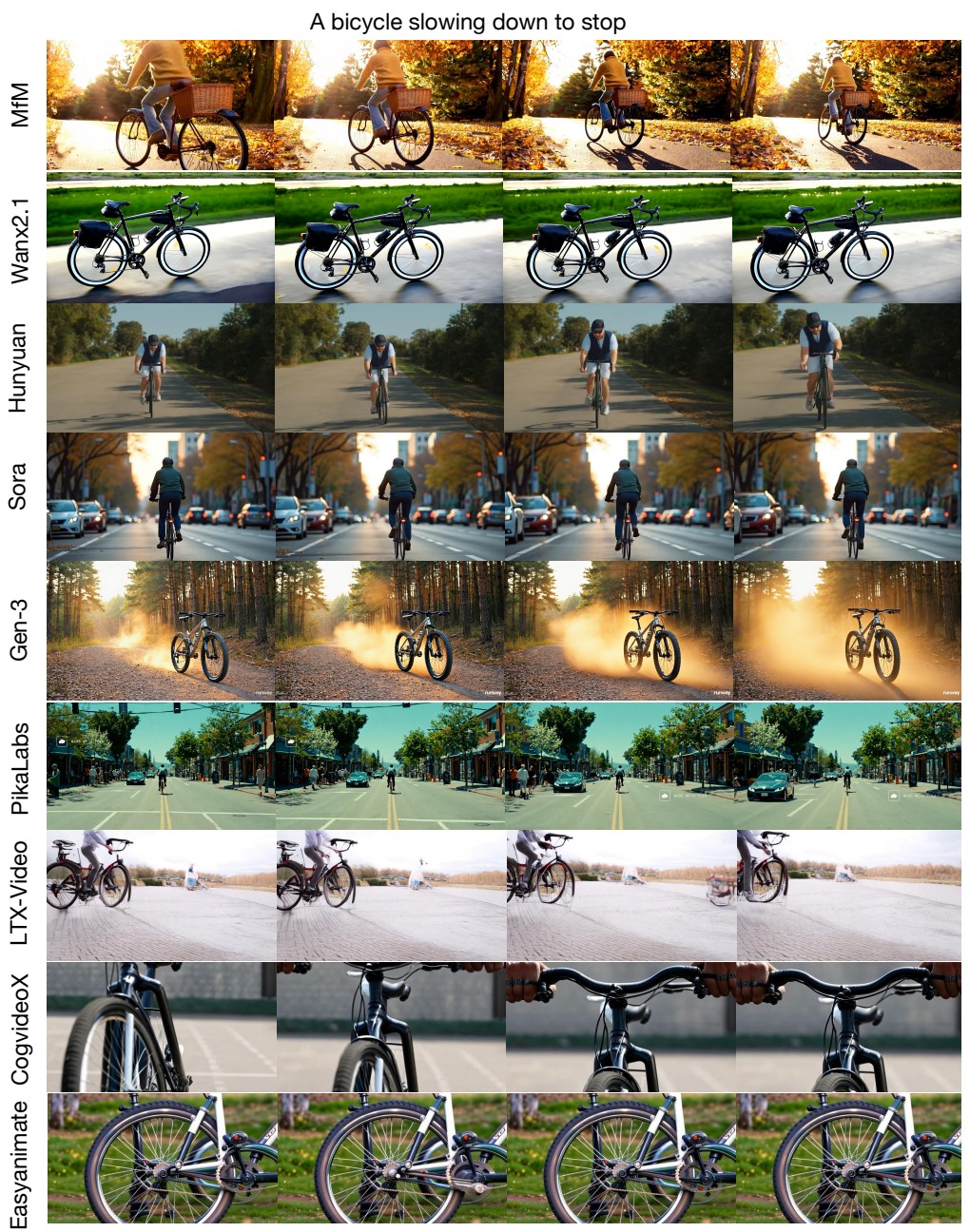

Figure 9: T2V generated videos with prompt "a bicycle slowing down to stop".

a sea turtle swimming in the ocean under the water.

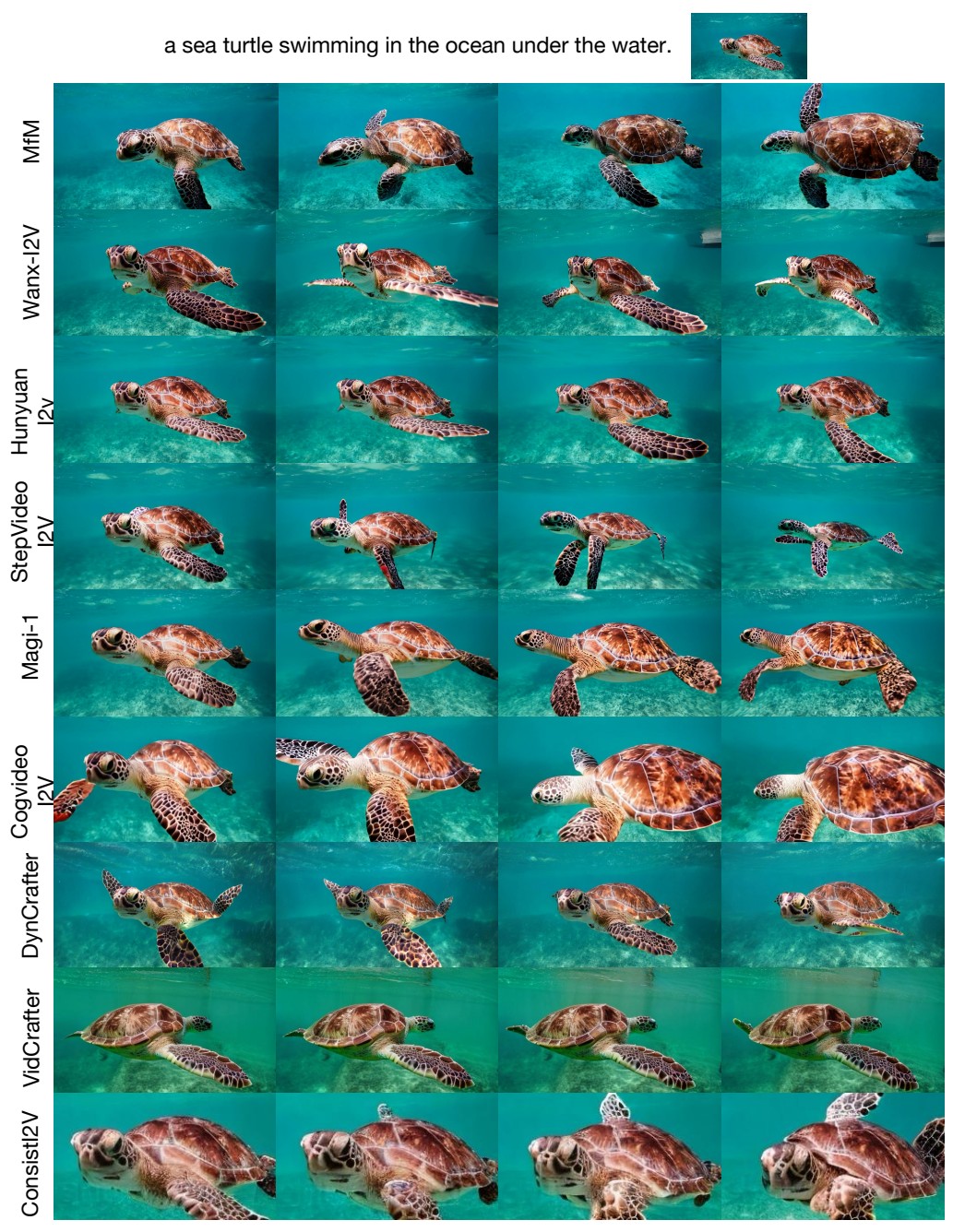

Figure 10: I2V generated videos with prompt "a sea turtle swimming in the ocean under the water".

a dog carrying a soccer ball in its mouth.

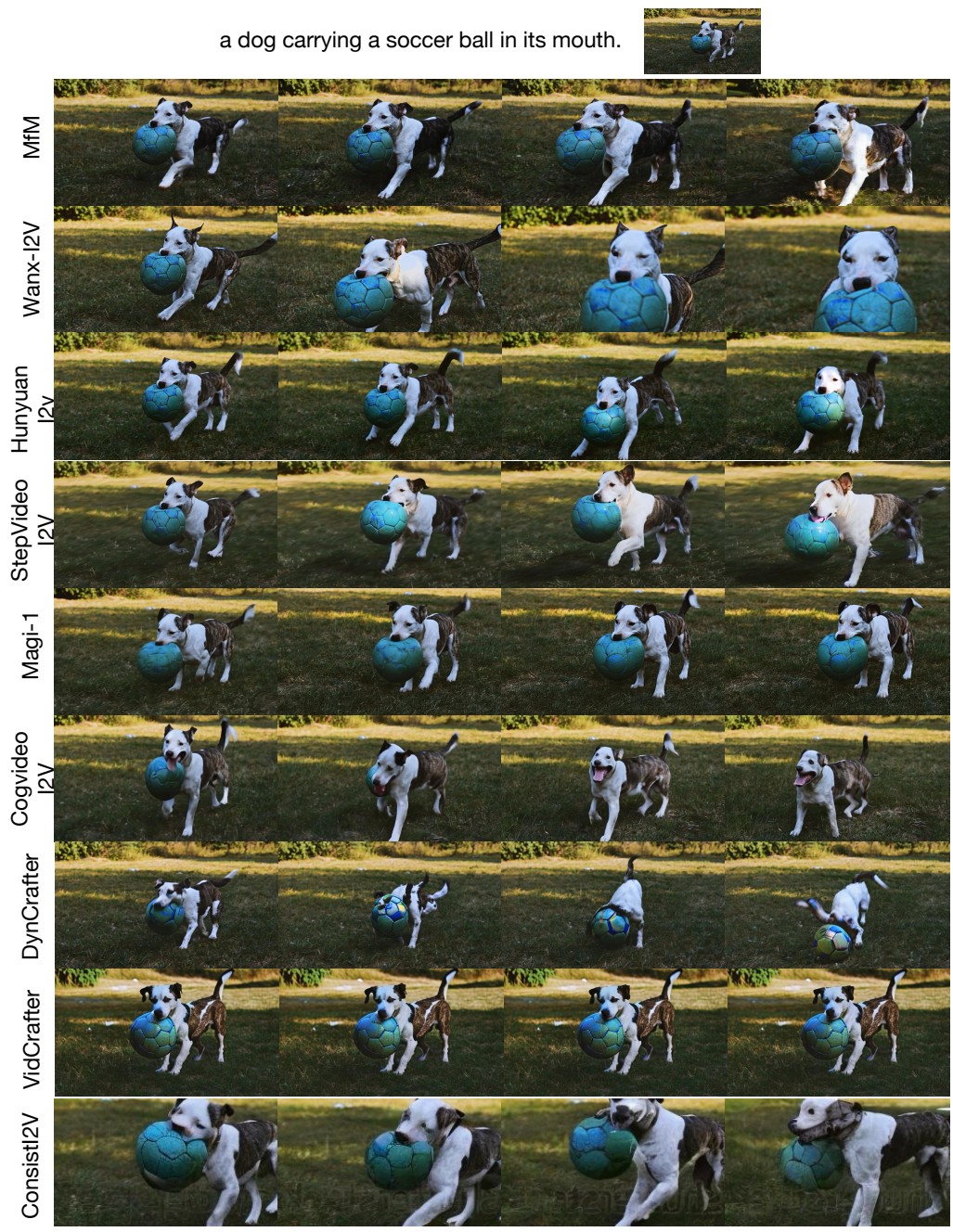

Figure 11: I2V generated videos with prompt "a dog carrying a soccer ball in its mouth".

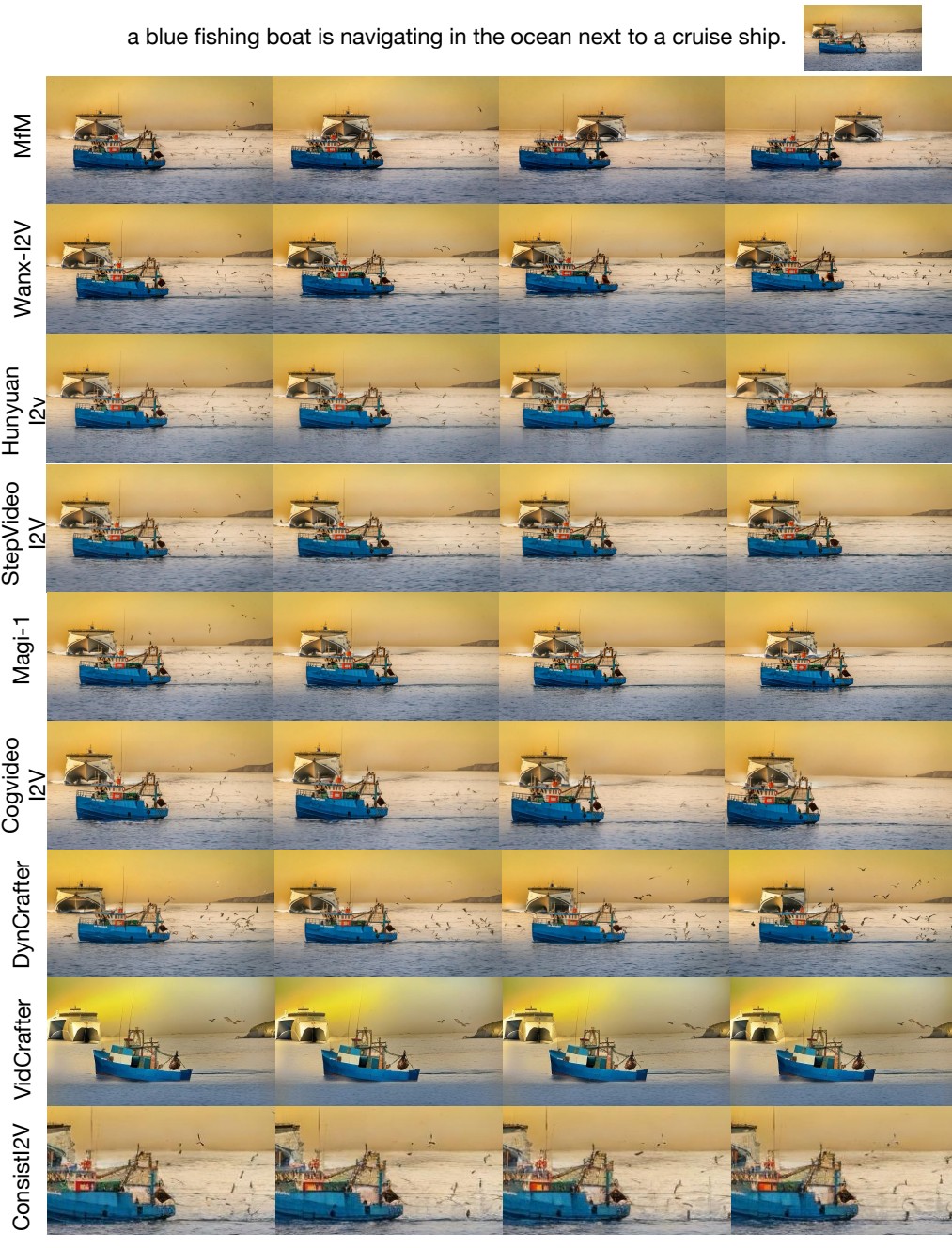

Figure 12: I2V generated videos with prompt "a blue fishing boat is navigating in the ocean next to a cruise ship".

a couple of horses are running in the dirt

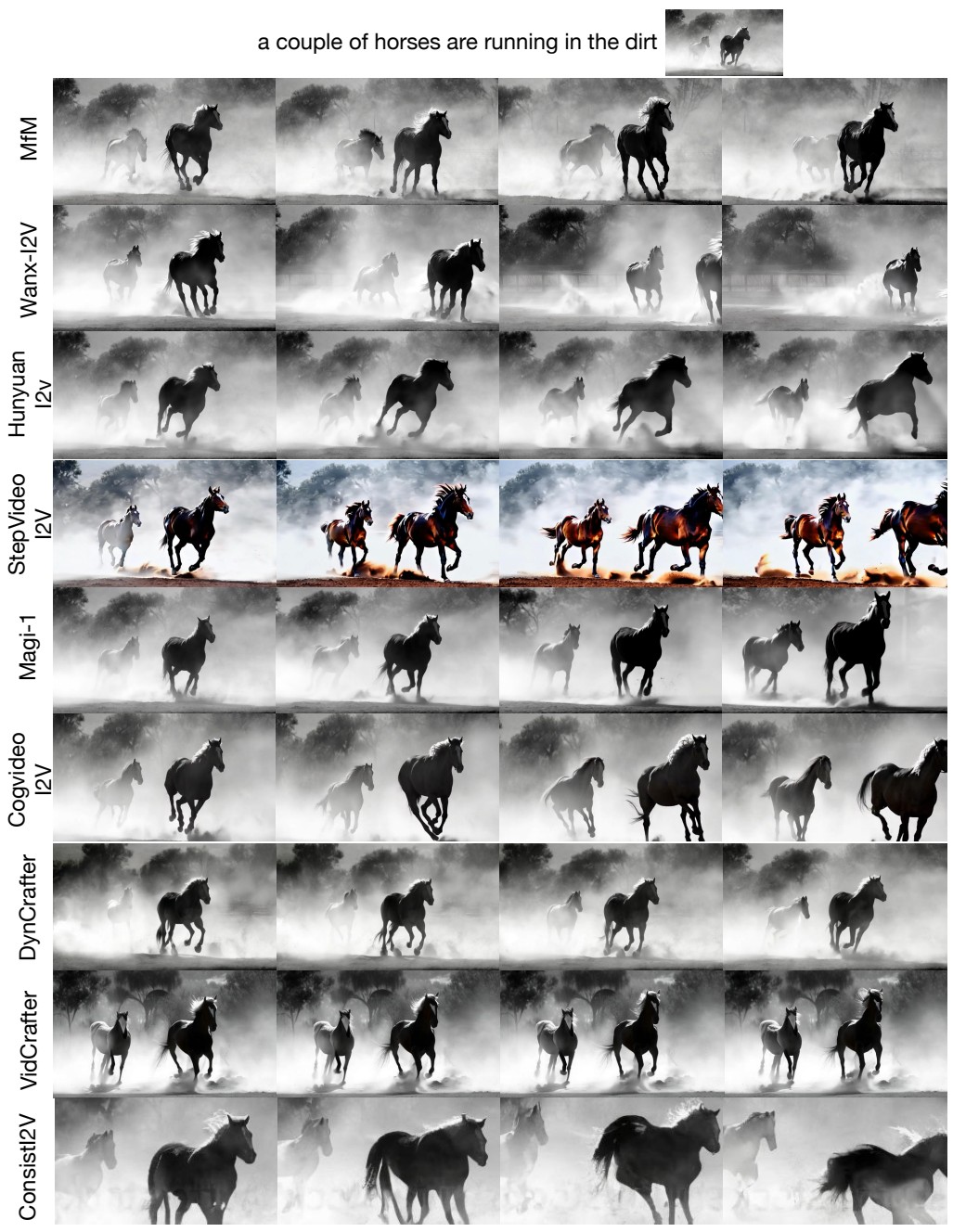

Figure 13: I2V generated videos with prompt "a couple of horses are running in the dirt".

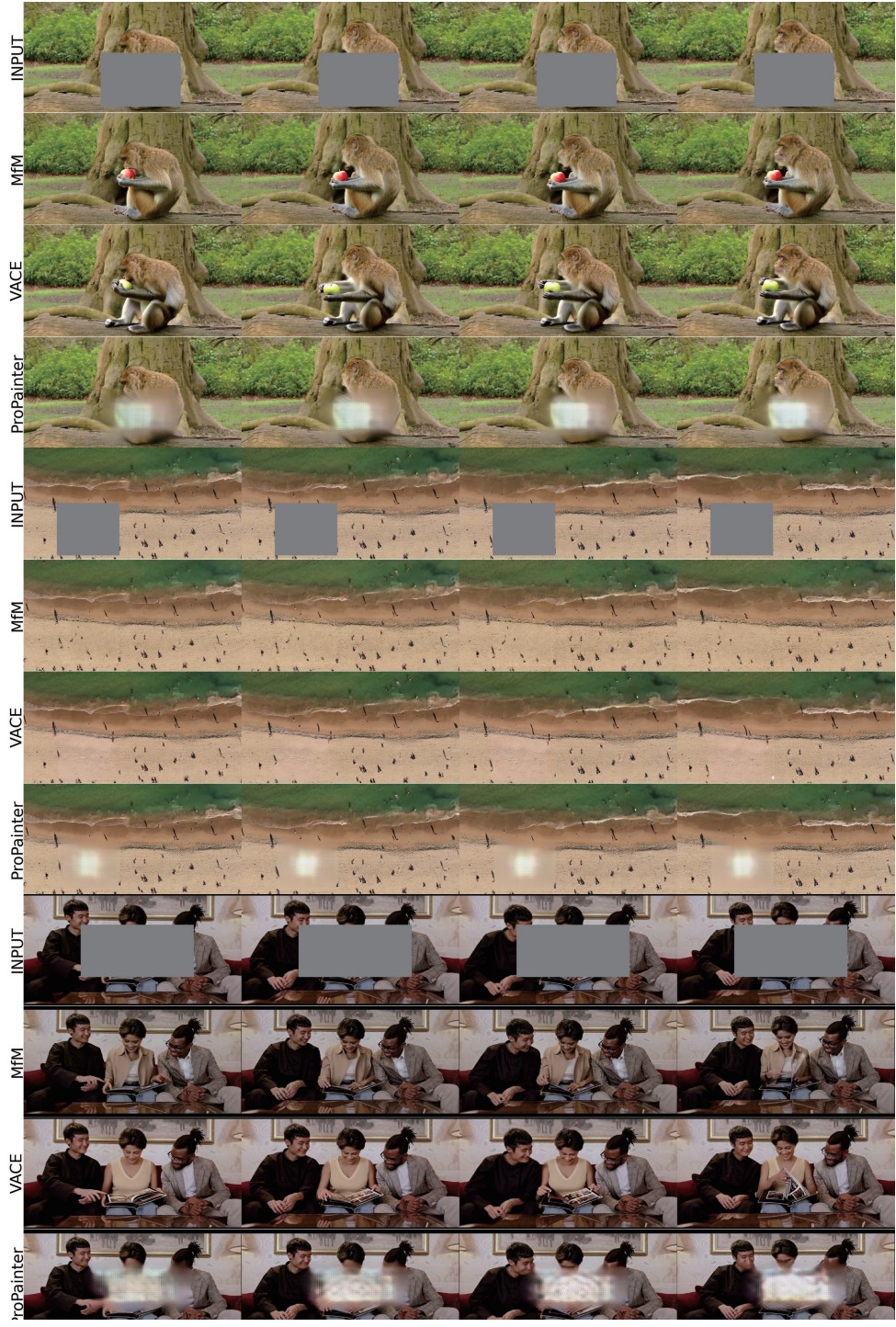

Figure 14: Visual comparison on task of video inpainting.

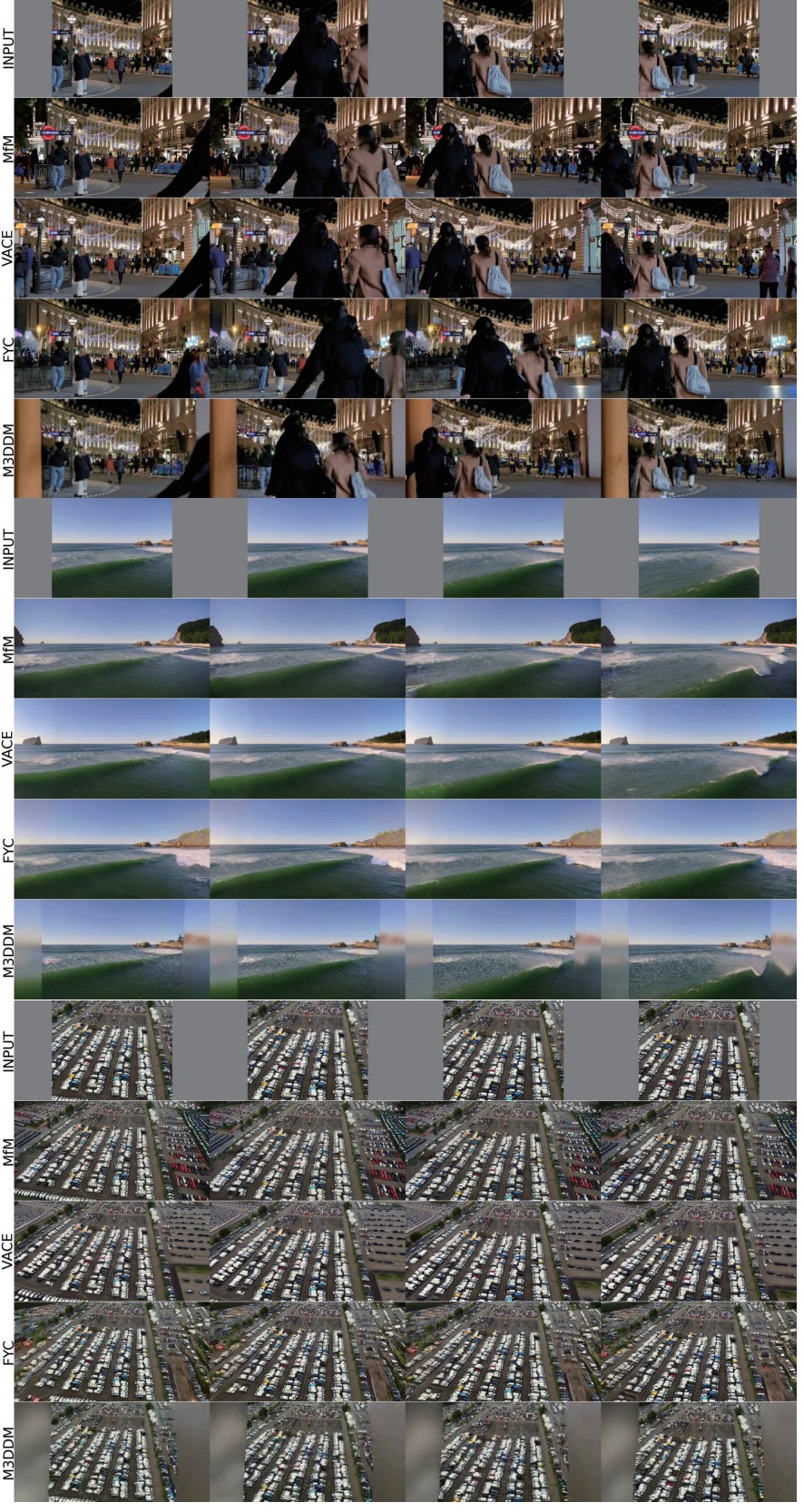

Figure 15: Visual comparison on task of video outpainting.

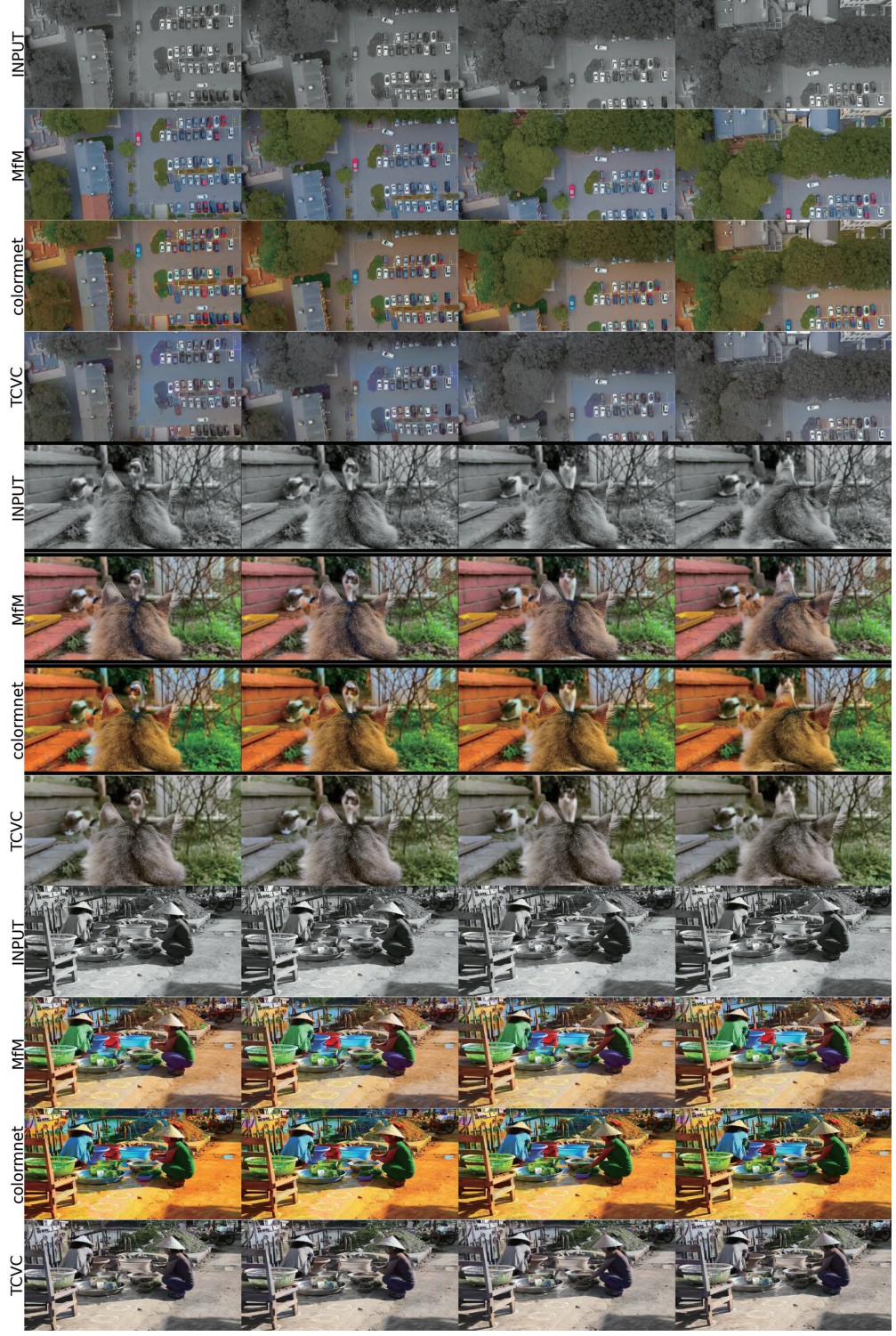

Figure 16: Visual comparison on task of video colorization.

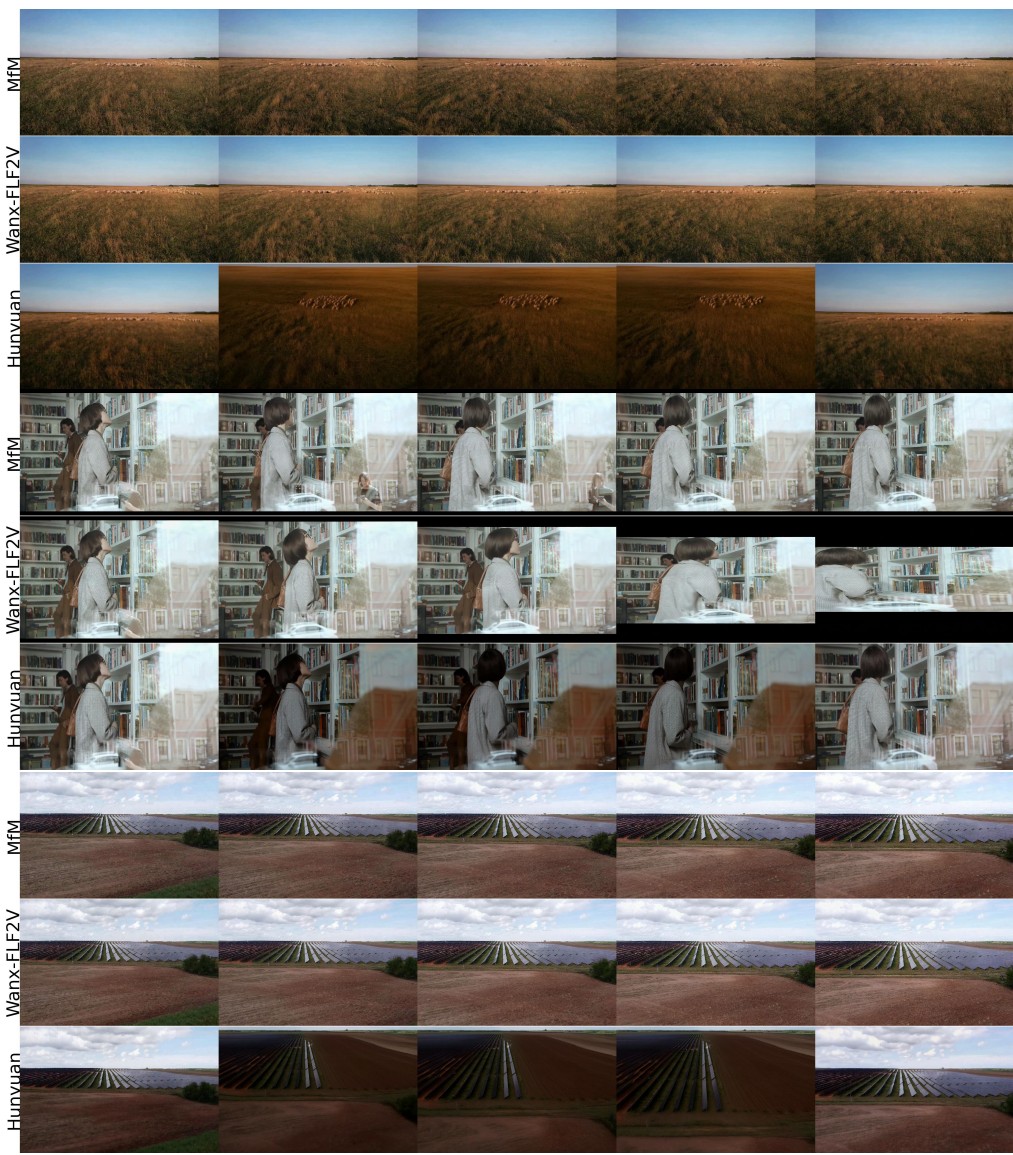

Figure 17: Visual comparison on task of first-last-frame to video.

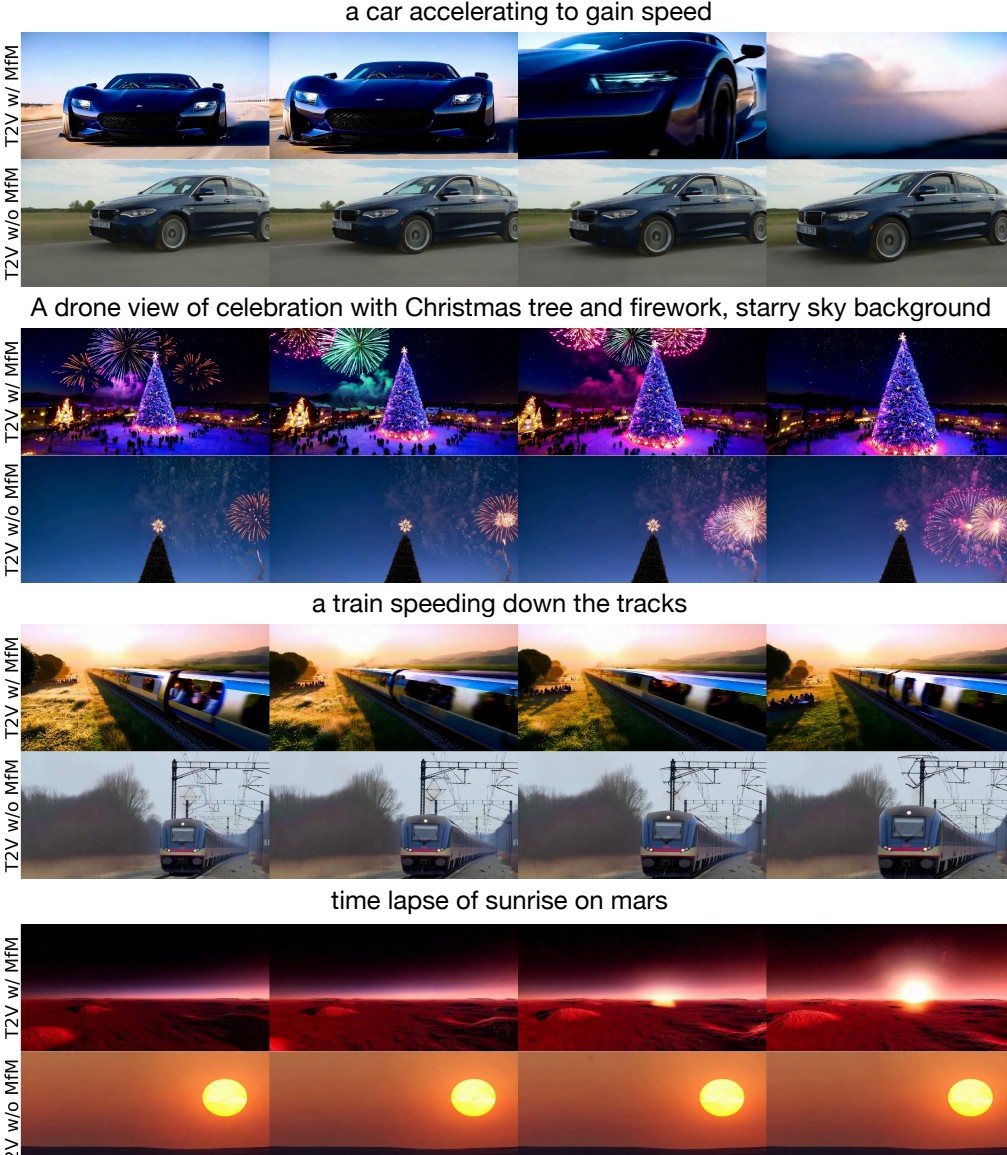

Figure 18: Ablation study on t2v task using MfM with/without multi-task training.

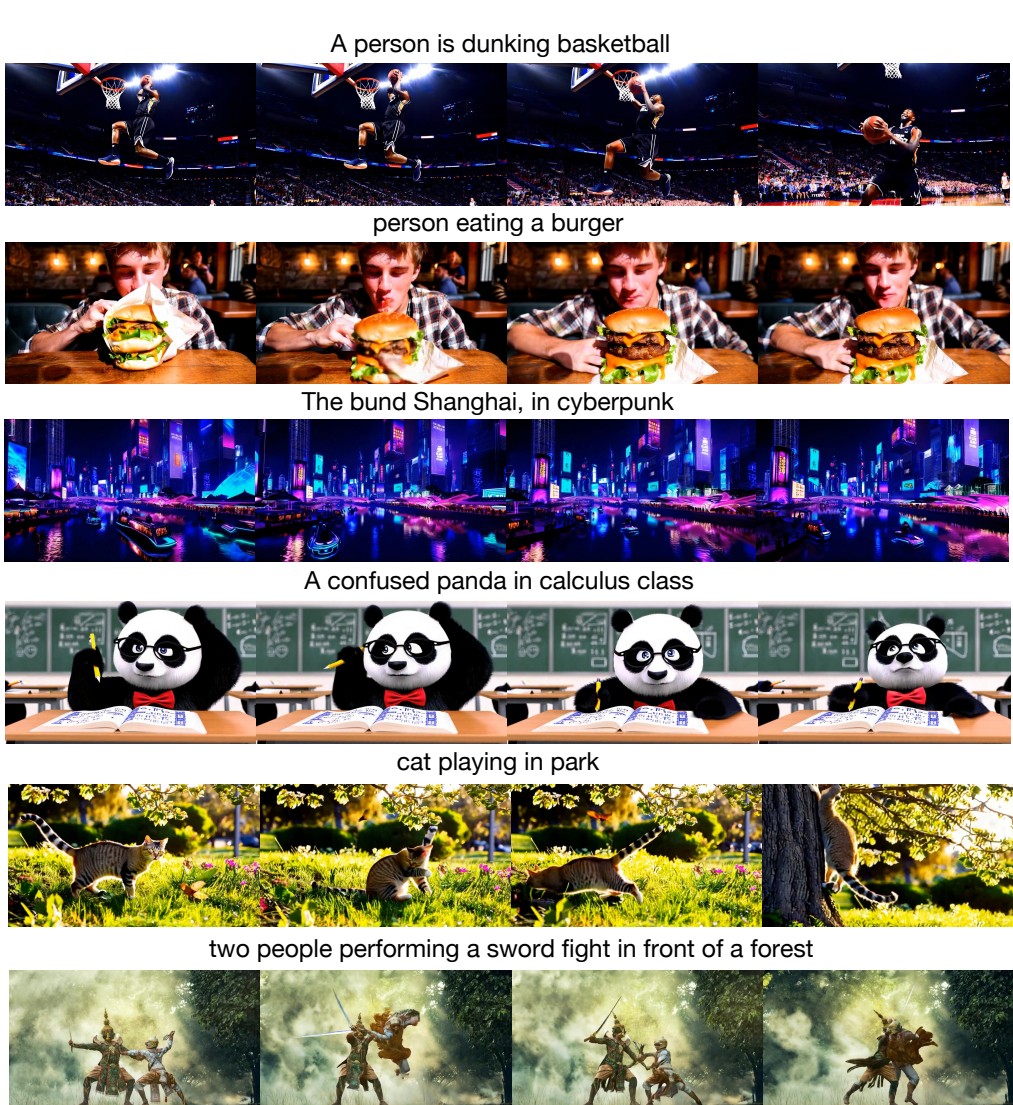

A person is dunking basketball

person eating a burger

The bund Shanghai, in cyberpunk

A confused panda in calculus class

cat playing in park

two people performing a sword fight in front of a forest

Figure 19: Failure cases of our MfM.

