# OpenReview forum: "Many-for-Many: Unify the Training of Multiple Video and Image Generation and Manipulation Tasks"
_ICLR.cc/2026/Conference — ICLR 2026 Poster_

### Official Review · Reviewer_7qTM · 2025-10-30

**Soundness:** 3
**Presentation:** 3
**Contribution:** 2
**Rating:** 6
**Confidence:** 4

**Summary:**

The authors propose a "many-for-many" framework, a unified diffusion model trained from scratch to handle numerous (10+) visual generation and manipulation tasks simultaneously (e.g., T2V, I2V, V2V). This approach avoids the high cost of training specialized foundation models. Key technical contributions include a lightweight adapter to unify diverse task conditions and a joint image-video learning strategy. The authors also integrate depth maps as a condition to improve 3D spatial awareness. The resulting models (8B and 2B parameters) demonstrate highly competitive performance, with the 8B version reportedly rivaling commercial systems.

**Strengths:**

- The core strength is the shift from single-task models to a single, unified framework. Training one model for over 10 tasks addresses a significant issue of model fragmentation and resource-intensive training.

- The use of a lightweight adapter to harmonize different task inputs, combined with a progressive joint image-video learning strategy, presents a novel and practical method for efficient multi-task learning.

**Weaknesses:**

- Sub-optimal Task-Specific Performance: The "jack-of-all-trades" approach, while versatile, does not appear to achieve state-of-the-art  performance on all individual downstream tasks. This suggests the joint training may be averaging performance rather than achieving the desired synergistic effect (where tasks mutually improve one another). This limitation might stem from the difficulty of balancing diverse, and potentially lower-quality, training datasets.

- The main technical contributions (a condition adapter and a joint training strategy) appear somewhat incremental. Given the significant undertaking of training an 8B model from scratch, one might expect a more fundamental innovation in the network architecture itself, moving beyond adapters toward a more deeply unified design to handle the "many-for-many" paradigm.

**Questions:**

I am curious about the decision to train the models (especially the 8B version) entirely from scratch, given the immense computational cost. Could the authors elaborate on the rationale for this choice over fine-tuning a strong, existing pre-trained T2V or I2V model (e.g., Wanx)? While leveraging such a model would presumably reduce training overhead and provide a robust baseline, I wonder if the authors found that training from scratch was necessary to effectively implement the "many-for-many" framework—perhaps because existing models are too architecturally specialized to accommodate the 10+ diverse tasks.

---

> ### Author Response · Authors · 2025-11-20
>
> We sincerely thank this reviewer for the recognition that our unified training framework addresses the issue of model fragmentation, and that the lightweight adapter combined with progressive joint training enhances multi-task learning. We also thank this reviewer for the insightful questions on sub-optimal task-specific performance and technical contributions. Our itemized responses to all of this reviewer's comments can be found as follows.
>
> **Weak 1: Sub-optimal Task-Specific Performance**
>
> Actually, our model achieves state-of-the-art performance on a few tasks (e.g., T2V, I2V) despite using significantly less data and a smaller network, but we agree with the reviewer that achieving SOTA across all downstream tasks is challenging. This difficulty arises from two factors: (1) the combinatorial complexity of jointly optimizing many heterogeneous tasks, and (2) resource constraints that prevent us from exhaustively tuning each task combination. Therefore, we prioritized improving the model’s core generation ability, which serves as the foundation for all other tasks.
>
> Regarding task synergy, we conducted controlled experiments demonstrating that certain tasks contribute positively to the model's performance. As shown in Table 6 of the main paper, mixing diverse tasks improves the base T2V performance, indicating that multi-task signals act as a form of regularization that enhances representation quality. Please note that, without our MfM training strategy, when training with a similar amount of data, the model cannot achieve state-of-the-art performance on specific tasks such as T2V and I2V, further demonstrating the effectiveness of our MfM strategy.
>
> On the data side, while we applied a detailed multi-stage filtering pipeline (as described in our responses to weak 1 of Reviewer 78YU), we acknowledge that there remains room for improvement in both quality and scale of the data. Strengthening the data construction pipeline—better quality assessment, higher-fidelity video captioning, and expanded editing data—is an important direction.
>
> Finally, although the unified model may not achieve optimal performance on every individual task, it provides a strong and versatile foundation. Downstream users can further fine-tune the model on their specific tasks to obtain task-specialized versions, leveraging the broad capabilities learned during mixed training.
>
> **Weak 2: Technical Contributions**
>
> We understand that one may expect a more sophisticated architectural design for a unified multi-task video model. However, the main goal of this work is to investigate whether we can use a simple framework and limited training data to train a model that can not only do multiple generation tasks, but also achieve competitive T2V performance. Finally, we adopt the architecture presented in the main paper, which employs many existing architectural components of current works together with a lightweight adapter.
>
> Despite its simplicity, this design, together with our mixed training paradigm, enables our 8B MfM model to achieve state-of-the-art results on several tasks (e.g., T2V, I2V, first–last frame to video, and video outpainting). Note that we achieve this using only 120M (40M HQ and 80M relatively LQ videos) video clips and 160M images, while the models such as Wanx and Hunyuan use billions of training samples. We agree that a more fine-grained architecture and task-aware conditioning mechanisms may further improve performance, which is a very good direction for future work.
>
> Actually, we have conducted ablation studies to compare different conditioning injection strategies—for example, concatenating conditioning features with video latents along the channel or sequence dimension. However, these approaches increase training and inference cost. In particular, sequence-wise concatenation doubles the effective token length (since the number of condition tokens equals the number of video tokens), leading to substantial attention overhead. Please also refer to our response to Weak 1 from Reviewer 5PWG for more discussion.
>
> We sincerely hope that our above explanations can address this reviewer's concern about the technical contributions of our work.

---

> ### Author Response · Authors · 2025-11-20
>
> **Question 1: The Decision to Train the Models Entirely from Scratch**
>
> This decision is actually determined by the goal of this work: We want to verify that **whether we can train a competitive video generation and manipulation model using only 120M video clips and 160M images as training data**. To validate this point, **we must train the models from scratch**, instead of finetuning existing pre-trained T2V or I2V models. Our results demonstrate that we can indeed train such a model, which can do more than 10 different generative tasks and show competitive T2V/I2V performance with Wanx or Hunyuan that are trained on billion-scale datasets.
>
> Another work, VACE [1], is actually finetuned on the pretrained T2V model Wanx, as recommended by this reviewer.
> We further conducted quantitative comparisons with VACE on overlapping tasks (see Table 5 in the main paper), and the results show that our model achieves better performance than their fine-tuned models on these tasks. Additionally, we performed the following experiment: starting from the original T2V model, we conducted 10K-step T2V-only training (Model 1) and multi-task training (Model 2, excluding VINP tasks). After training, we fine-tuned both models on the new VINP task and compared their adaptation performance. The results, shown in the following table, demonstrate that the multi-task model adapts more effectively to the new task, highlighting the benefits of our unified multi-task pretraining approach (We have updated the corresponding results in Appendix Section A.15).
>
> | Method             | FVD↓  | PSNR↑ | SSIM↑    | LPIPS↓   | Aesthetic | Imaging Quality | Dynamic | Scene  | Consistency | Temporal  |
> |--------------------|---------|---------|--------|--------|--------|----------|---------|--------|---------|--------|
> |Model 1 | 758.894 | 28.740 | 0.699 | 0.296 | 0.5284 | 0.7201 | 0.9940 | 0.2258 | 0.9894 |
> |Model 2 | 740.026 | 29.113 | 0.709 | 0.286 | 0.5339 | 0.7347 | 0.9939 | 0.2281 | 0.9890 |
>
> [1] Jiang Z, Han Z, Mao C, et al. Vace: All-in-one video creation and editing[J]. arXiv preprint arXiv:2503.07598, 2025.

---

### Official Review · Reviewer_syLa · 2025-10-31

**Soundness:** 3
**Presentation:** 3
**Contribution:** 3
**Rating:** 4
**Confidence:** 4

**Summary:**

This paper proposes a unified diffusion transformer framework that jointly trains multiple visual generation and manipulation tasks, e.g., T2V, I2V, colorization, inpainting, super-resolution. The key contribution is a lightweight conditional adapter that unifies heterogeneous inputs (text, depth, masks, pixels) into a shared latent space, enabling multi-task learning through flow-matching and progressive image–video co-training. Experiments show that their method achieves competitive or superior results to current models.

**Strengths:**

The paper targets on unifying video generation and manipulation tasks. The adapter design is practical supporting seamless conditioning across modalities. The multi-task joint training strategy is demonstrated to be able to improve both performance and data efficiency. Ablation studies clearly show the benefits of multi-task learning and depth conditioning.

**Weaknesses:**

1. The architectural novelty of the proposed framework is limited. While the paper presents a unified system, its backbone design largely follows existing diffusion transformer paradigms such as SD3. The main contribution lies in integrating known components, i.e. flow matching, 3D attention, and adapter-based conditioning, into a unified training pipeline, rather than introducing new modeling framework.
2.  Although the results are promising, the evaluation is based on relatively limited datasets (mainly VBench and a small in-house MfM-benchmark with 480 samples across 16 tasks). This raises concerns about the generality and robustness of the model’s performance, especially on open-domain or long-duration video generation benchmarks.
3. While multi-task joint training is central to the paper, there is no quantitative or qualitative study on whether learning one task negatively affects another. Understanding these interactions would strengthen the multi-task learning claims.

**Questions:**

1. How is the adapter output integrated into the DiT pipeline additively in latent space or concatenated as condition tokens?
2. Are task sampling probabilities fixed or dynamically adjusted during training?
3. What quantitative gains do Q–K normalization and 3D RoPE bring to training stability?

---

> ### Author Response · Authors · 2025-11-20
>
> We sincerely thank this reviewer for the recognition on the SOTA performance and high data efficiency enabled by our multi-task joint training strategy. We also thank this reviewer for the insightful questions on architectural design, robustness on open-domain benchmarks and multi-task interactions. Our itemized responses to all of this reviewer's comments can be found as follows.
>
> **Weak 1: Limited Architectural Novelty**
>
> Thanks for the question. In this work, our goal is not to introduce an entirely new architectural primitive. Instead, our goal is to demonstrate whether we can train a unified many-for-many video generation framework, which can not only support 10+ heterogeneous generation and editing tasks, but also achieve competitive performance using limited training data (40M HQ and 80M relatively LQ videos and 160M images, compared with billions of data used in other models such as Wanx, Hunyuan) and computing budget.
>
> With the above goal, a central contribution of our work is that we show that diverse visual generation and editing tasks can be jointly optimized in a shared latent and conditioning space. As evidenced by Table 6 of the main paper, the introduction of additional editing tasks leads to measurable gains in T2V performance, even though these tasks are not designed specifically for T2V. This highlights a key finding: multi-task co-training serves as an implicit form of regularization, enabling the model to learn richer video representations and excel beyond what a pure T2V setting could achieve with the same training budget.
>
> On the architectural side, our adapter is intentionally designed to be as simple as possible. Given the heavy computational footprint of the DiT backbone, we designed an adapter that is both task-agnostic and cost-neutral. The 5-layer convolutional adapter injects conditioning information without increasing attention length or adding meaningful computational overhead, allowing the model to scale gracefully as more tasks are included. (For the computational overhead, please also refer to our responses to Weak 1 of Reviewer 5PWG for more discussions.)
> However, we agree that a more fine-grained architectural design may further improve performance, which is a very good direction for future work.
>
> We sincerely hope that our above explanations can address this reviewer's concern about the architectural novelty of our work.
>
> [1] Polyak A, Zohar A, Brown A, et al. Moviegen: A cast of media foundation models[J].
>
> [2] Google. Veo: a text-to-video generation system, 2025.
>
> [3] Chen S, Ge C, Zhang Y, et al. Goku: Flow based video generative foundation models[C].

---

> ### Author Response · Authors · 2025-11-20
>
> **Weak 2: Results on More Benchmarks**
>
> As suggested by this reviewer, we adopt another widely used benchmark — the MovieGen Benchmark [1] released by Meta (hereafter referred to as the MovieGen Benchmark) — for further evaluation. This benchmark provides broader coverage across key evaluation dimensions and includes diverse motion categories (i.e., high, medium, and low motion prompts). It has also been adopted by recent state-of-the-art works such as Veo 3 [2] and Goku [3].
> Since the MovieGen Benchmark does not provide official evaluation metrics, we employ the same set of video quality evaluation metrics used in VBench to measure the performance of different models. The results of open-sourced models are presented in the following table (We have updated the corresponding results in Appendix Section A.12) :
>
> | Method             | Aesthetic  | Imaging Quality | Dynamic    | Temporal   | Consistency | Avg. Rank. |
> |--------------------|---------|---------|--------|--------|--------|----------|
> |MfM (8B) | 0.6136 | *0.6887* | **0.7188** | 0.2615 | *0.2615* | **2.2** |
> |Wanx (14B) | **0.6428** | **0.7265** | 0.2969 | 0.2474 | 0.2474 | 3.4 |
> |Hunyuan (13B) | 0.6044 | 0.6483 | *0.5469* | *0.2651* | **0.2651** | *2.4* |
> |Opensora (11B) | *0.6311* | 0.6180 | 0.5156 | **0.2688** | 0.2588 | 2.6 |
> |Cogvideo (5B) | 0.5634 | 0.6104 | 0.4844 | 0.2452 | 0.2452 | 4.8 |
> |EasyAnimate (12B) | 0.5439 | 0.5697 | 0.4322 | 0.1375 | 0.1375 | 6.2 |
> |LTX-video (0.98B) | 0.5087 | 0.5603 | 0.3438 | 0.2063 | 0.2063 | 6.8 |
>
> The results on the MovieGen benchmark further demonstrate the advantages of our MfM. In particular, MfM (8B) achieves the best performance on video dynamics and the second-best results on image quality and overall consistency, closely matching or even surpassing those much larger models such as Wanx (14B), Hunyuan (13B), and OpenSora (11B). Note that our **MfM model achieves this performance by training on only 120M video clips and 160M images**, far less than those models like Wanx and Hunyuan, which are trained on billion-scale datasets.
>
> Regarding long-duration video generation, our model is primarily trained on clips of 97 frames. Extending the temporal window significantly increases the computational and memory cost during training, and thus long-duration generation is currently beyond the intended scope of this work. A promising direction is to adapt our MfM framework to a streaming or chunk-wise generation pipeline, which would enable arbitrarily long videos. We consider this as an important extension in future work.

---

> ### Author Response · Authors · 2025-11-20
>
> **Weak 3: Interactions between Different Tasks**
>
> Given the large number of possible task combinations, it is infeasible to conduct an exhaustive ablation over all of them. Therefore, we select a subset of representative tasks (T2V, VINP, FLF2V, VColor) to study task interactions. In particular, starting from a T2V-only checkpoint, we evaluate mixing strategies including: T2V only, T2V + any single task, T2V + any two tasks, and T2V + all selected tasks.
> The evaluation results on VBench-T2V are summarized below (we have updated the corresponding results in Appendix Section A.13):
>
> | Method             | Motion  | Dynamic | Aes    | Img.   | Object | Mul.Obj. | Spatial | Scene  | Appear. | Temp.  | Consist. |
> |--------------------|---------|---------|--------|--------|--------|----------|---------|--------|---------|--------|----------|
> |T2V                   |0.9915    |0.6556    |0.5931    |0.5448    |*0.8869*    |0.5457    |*0.6112*    |0.5065    |0.2214    |0.2379    |0.2520 |
> |T2V+VCOLOR            |0.9892    |0.6111    |0.5881    |0.5484    |0.8339    |0.5122    |0.4326    |0.5000    |0.2317    |0.2406    |0.2519 |
> |T2V+VINP              |0.9887    |0.6944    |0.5770    |0.5402    |0.8703    |0.5655    |0.5714    |0.4833    |*0.2333*    |0.2288    |0.2474 |
> |T2V+FLF2V             |0.9911    |0.7778    |*0.5930*    |0.5437    |**0.8932**    |**0.6006**    |0.5098    |0.4935    |0.2295    |0.2415    |0.2464 |
> |T2V+FLF2V+VCOLOR      |0.9846    |0.7639    |0.5743    |0.5725    |0.8623    |0.5358    |**0.6515**    |*0.5291*    |0.2298    |*0.2419*    |0.2551 |
> |T2V+VINP+FLF2V        |**0.9925**    |*0.8472*    |**0.5947**    |*0.5808*    |0.9090    |*0.5983*    |0.5261    |**0.5356**    |0.2291    |**0.2421**    |*0.2551* |
> |T2V+VINP+VCOLOR       |0.9870    |0.7778    |0.5816    |0.5785    |0.8576    |0.5816    |0.5150    |0.5007    |**0.2347**    |0.2333    |0.2445 |
> |T2V+FLF2V+VCOLOR+VINP |*0.9922*    |**0.8750**    |0.5911    |**0.5853**    |0.8861    |0.5760    |0.5851    |0.4969    |0.2289    |0.2372    |**0.2553** |
>
> As shown in the table, in most cases, when both tasks improve performance on certain metrics, incorporating them into the mixed training pipeline also brings benefits (e.g., dynamic, appearance). However, when one task improves performance while another negatively affects it, the final mixed results vary across metrics. For example, for object and multiple object metrics, integrating FLF2V leads to significant gains. When further combining it with VCOLOR, the mixed model still improves upon the T2V baseline, but the magnitude of improvement narrows because VCOLOR has a mild negative effect on single-object and multi-object generation accuracy.
> Conversely, mixing VINP and VCOLOR with T2V can degrade performance on temporal style, though VCOLOR alone improves general generation quality.
>
> Interestingly, we also observe cases where two individually harmful tasks produce unexpected improvements when combined. For instance, VINP and FLF2V each weaken some metrics, but together, they significantly improve scene and overall consistency. We attribute this to the complementary regularization effects they impose on holistic video understanding: although VINP primarily focuses on spatial completion and FLF2V on temporal completion, each task alone may bias optimization in an unbalanced direction, whereas their combination better constrains the model and leads to improved global metrics like overall consistency.
>
> Finally, when all tasks are included in training, the model achieves substantial gains on most metrics (e.g., motion, dynamic, image quality, appearance, overall consistency). Some metrics drop slightly compared to T2V-only training due to the absence of regularization from tasks that specifically benefit them. For example, as shown in Table 6 of the main paper, metrics such as aesthetics and scene benefit greatly from incorporating video-extension and first-last-clip-to-video tasks.
>
> However, it is difficult to accurately evaluate the effect of each individual task and all possible combinations, given the enormous combinatorial space and the complex interactions among tasks.
> Therefore, throughout this work, we focus on assessing the overall performance of the model under mixed-task training—examining whether the model can simultaneously learn 10+ editing capabilities while leveraging their interactions to improve video generation ability.

---

> ### Author Response · Authors · 2025-11-20
>
> **Question 1: Integration of Adapter Output into the DiT Pipeline**
>
> The conditioning information is injected by passing it through our adapter, which produces a feature map with the same shape as the video latent. We also experimented with alternative concatenation-based designs—concatenating along the channel dimension or along the sequence dimension. These approaches yielded similar performance, confirming that the choice of injection mechanism is not the primary bottleneck. However, both of these two concatenation strategies introduce non-trivial computational and memory overhead (e.g., increased projection parameters or longer token sequences).
>
> In contrast, the additive formulation is extremely lightweight—its extra computational cost is negligible—while achieving comparable performance. For this reason, we adopt the addition-based injection in our final design.
>
> **Question 2: Task Sampling Probabilities**
>
> Because dynamically adjusting task ratios during training will introduce additional complexity and often lead to instability, we keep the task proportions fixed throughout the entire training process, which achieves promising performance.
>
> **Question 3: Quantitative Gains of Q–K Normalization and 3D RoPE**
>
> To better clarify the contribution of Q–K Normalization and 3D Rotary Position Embedding (3D RoPE), we performed an ablation study by finetuning a 2B-parameter checkpoint for an additional 10K training steps under three settings: (1) full model, (2) removing Q–K Norm, and (3) removing 3D RoPE. We report the results on the VBench-T2V benchmark in the table below (We have updated the corresponding results in Appendix Section A.14):
>
> | Method             | Motion  | Dynamic | Aes    | Img.   | Object | Mul.Obj. | Spatial | Scene  | Appear. | Temp.  | Consist. |
> |--------------------|---------|---------|--------|--------|--------|----------|---------|--------|---------|--------|----------|
> |MfM-baseline | 0.9922 | 0.8889 | 0.5911 | 0.5853 | 0.8861 | 0.5160 | 0.5851 | 0.4869 | 0.2289 | 0.2341 | 0.2483  |
> |MfM (w/o Q-K Norm) | NA | NA | NA | NA | NA | NA | NA | NA | NA | NA | NA |
> |MfM (w/o RoPE) | 0.9659 | 0.8472 | 0.3149 | 0.3628 | 0.0973 | 0.0000| 0.0089 | 0.0022 | 0.2200 | 0.0423 | 0.0584 |
>
> In our experiments, the model consistently collapses after ~3K steps once Q–K Normalization is removed. This collapse manifests as exploding attention activations and rapidly diverging losses, blocking further training. This confirms that Q–K Norm is critical for stabilizing large-scale DiT-based video generation models, especially under our multi-task many-for-many training regime where diverse conditioning signals create additional gradient variance.
>
> Unlike Q–K Norm, removing 3D RoPE does not cause training divergence; however, it leads to substantial degradation across all VBench metrics. The drop is particularly severe for spatial–semantic metrics such as Object, Multiple Objects, Spatial, and Scene. Without 3D RoPE, the model frequently fails to place objects in correct spatial locations or maintain consistent geometry throughout the video, resulting in near-zero performance on these categories. This demonstrates that 3D RoPE is crucial for modeling the joint spatial–temporal structure of video tokens.

---

> > ### Comment · Reviewer_syLa · 2025-11-24
> >
> > Thanks for the response, I raised the score to 6. Good luck!

---

> > > ### Author Response · Authors · 2025-11-24
> > >
> > > We sincerely thank you for your positive feedback and your recognition of our work!
> > >
> > > Best regards,
> > >
> > > Authors of paper #2736

---

### Official Review · Reviewer_78Yu · 2025-10-31

**Soundness:** 3
**Presentation:** 3
**Contribution:** 3
**Rating:** 6
**Confidence:** 2

**Summary:**

This paper introduces MfM (Many-for-Many), a unified framework for visual generation and manipulation. The core contribution is a single diffusion transformer model trained from scratch to handle over ten distinct tasks. Task unification is achieved via lightweight adapters for conditional inputs and by appending task-specific names to text prompts to guide the model’s behavior. The authors demonstrate state-of-the-art performance on the VBench benchmark, outperforming several open-source and commercial models.

**Strengths:**

1. The proposed MfM framework is both simple and elegant. The use of a unified adapter for diverse 3D conditions (including pixel data, depth, and masks), combined with task-name conditioning in the text prompt, represents an effective and scalable solution. This approach successfully unifies a wide range of visual generation and manipulation tasks within a single model, eliminating the need for task-specific fine-tuning.

2. The model demonstrates empirically strong performance, achieving the highest average rank on both the VBench-T2V and VBench-I2V benchmarks. The results are well-validated and indicate the robustness of the proposed method.

**Weaknesses:**

1. MfM is trained using proprietary data, but the authors do not clearly delineate the extent to which the model’s performance is attributable to the MfM framework itself versus the use of high-quality proprietary data. This ambiguity significantly limits the reference value of this work for the broader research community.

2. Regarding the composition of the training data, the authors mention that the sampling probability for T2I, T2V, and I2V tasks is three times higher than for other tasks. It is unclear whether this is an empirical choice or if there are additional ablation studies or pilot experiments to support this decision.

**Questions:**

As an academic paper, providing more detailed transparency regarding the training data would greatly enhance the value of this work. I am particularly interested in whether the authors can more clearly disentangle and demonstrate the respective contributions of the data and the proposed method to the overall performance.

---

> ### Author Response · Authors · 2025-11-20
>
> We sincerely thank this reviewer for the recognition on the simplicity and elegance of our framework, as well as its strong performance. We also thank this reviewer for the insightful questions on data construction and task sampling ratios. Our itemized responses to all of this reviewer's comments can be found as follows.
>
> **Weak 1 & Question 1: Details and Roles about Training Data**
>
> Regarding our training data, over 70% of them are collected from publicly available sources, including Panda70M [1], Koala36M [2], InternVid [3], OpenVid [4], and WebVid [5], complemented by a small portion of proprietary data. For images, the primary source is LAION-5B [6]. To ensure data quality, we adopt a multi-stage filtering pipeline (We have updated the corresponding
> results in Appendix Section A.10):
>
> - Video Segmentation: We first apply PySceneDetect for coarse scene boundaries. Then, we extract frame-level features using DINOv2 [7], compute inter-frame similarity, and further split clips at low-similarity points. Videos shorter than 2 seconds are removed.
> - Video Quality Filtering: Each segmented clip is evaluated along several dimensions: 1) basic metadata (FPS, resolution, bitrate) extracted directly from video; 2) average aesthetic score using a pretrained aesthetic model; 3) overlay text ratio via a pretrained OCR model; 4) watermark detection through a dedicated watermark model; 5)for motion quality, we compute optical flow using RAFT, then filter out clips with insufficient motion.
> - Semantic Content Filtering: To identify and remove potential low-quality or undesirable content, we employ a fine-tuned VideoLLaMA3 [8] model to detect unsafe content, low-light or blurry scenes, overexposed frames, black borders, abrupt perspective shifts, and static-image animations.
> - Video Captioning: For caption generation, we use Tarsier2 [9], prompting it to produce two complementary captions: a short global summary and a long, detailed description. These two captions are merged to form the final caption for each clip.
>
> As for the contribution of our training data to the model, since existing open-source models are trained on different datasets with data volumes much larger than ours, it is difficult to disentangle the exact contribution of dataset quality versus training paradigm when making cross-model comparisons. Therefore, we validate the effectiveness of our proposed mixed-training paradigm through controlled experiments. As shown in Table 6 of the main paper, we conduct a fair comparison under the same training budget: training the model solely on T2V data, versus training the model with our many-for-many mixed-task paradigm.
>
> We can see that across all evaluation metrics, the model trained with our mixed-task paradigm consistently outperforms the model trained with pure T2V data on T2V benchmarks. We attribute this improvement to multi-task regularization, where diverse supervisory signals encourage the model to learn richer and more generalizable video representations.
> This provides strong evidence that the performance gain is largely brought by the proposed mixed-training paradigm, rather than by dataset differences.
>
> [1] Chen T S, Siarohin A, Menapace W, et al. Panda-70m: Captioning 70m videos with multiple cross-modality teachers[C]//Proceedings of the IEEE/CVF Conference on Computer Vision and Pattern Recognition. 2024: 13320-13331.
>
> [2] Wang Q, Shi Y, Ou J, et al. Koala-36m: A large-scale video dataset improving consistency between fine-grained conditions and video content[C]//Proceedings of the Computer Vision and Pattern Recognition Conference. 2025: 8428-8437.
>
> [3] Wang Y, He Y, Li Y, et al. Internvid: A large-scale video-text dataset for multimodal understanding and generation[J]. arXiv preprint arXiv:2307.06942, 2023.
>
> [4] Nan K, Xie R, Zhou P, et al. Openvid-1m: A large-scale high-quality dataset for text-to-video generation[J]. arXiv preprint arXiv:2407.02371, 2024.
>
> [5] Bain M, Nagrani A, Varol G, et al. Frozen in time: A joint video and image encoder for end-to-end retrieval[C]//Proceedings of the IEEE/CVF international conference on computer vision. 2021: 1728-1738.
>
> [6] Schuhmann C, Beaumont R, Vencu R, et al. Laion-5b: An open large-scale dataset for training next generation image-text models[J]. Advances in neural information processing systems, 2022, 35: 25278-25294.
>
> [7] Oquab M, Darcet T, Moutakanni T, et al. Dinov2: Learning robust visual features without supervision[J]. arXiv preprint arXiv:2304.07193, 2023.
>
> [8] Zhang B, Li K, Cheng Z, et al. Videollama 3: Frontier multimodal foundation models for image and video understanding[J]. arXiv preprint arXiv:2501.13106, 2025.
>
> [9] Yuan L, Wang J, Sun H, et al. Tarsier2: Advancing large vision-language models from detailed video description to comprehensive video understanding[J]. arXiv preprint arXiv:2501.07888, 2025.

---

> ### Author Response · Authors · 2025-11-20
>
> **Weak 2: Ablation Study on Sampling Probability**
>
> We chose to assign a 3× higher sampling probability to the basic generation tasks for two main reasons. First, these three tasks (T2V, I2V, and T2I) represent the core generation capabilities commonly required in practical scenarios. Unlike editing tasks, which provide strong and explicit conditioning signals, these basic generation tasks rely on weaker supervision and are therefore significantly harder to optimize. Allocating additional sampling probability ensures that the backbone's generative ability can be sufficiently strengthened during pretraining.
>
> Second, we conducted an ablation study to investigate how different sampling ratios affect model performance. Specifically, we compared four settings: 1) Equal sampling probability across all tasks; 2) 2x sampling probability for the three basic tasks; 3) 3x sampling probability for the three basic tasks; and 4) 4x sampling probability for the three basic tasks.
>
> The evaluation results on VBench-T2V are shown in the following table (We have updated the corresponding results in Appendix Section A.11).
>
> | Method             | Motion  | Dynamic | Aes    | Img.   | Object | Mul.Obj. | Spatial | Scene  | Appear. | Temp.  | Consist. |
> |--------------------|---------|---------|--------|--------|--------|----------|---------|--------|---------|--------|----------|
> |1x | *0.9865* | 0.6528 | 0.5822 | 0.5493 | *0.8726* | **0.6395** | **0.6026** | 0.4789 | 0.2254 | 0.2299 | **0.2507**  |
> |2x | 0.9664 | *0.8750* | 0.5520 | 0.5420 | 0.7642 | 0.3361 | 0.4578 | **0.4942** | *0.2285* | 0.2272 | *0.2495*  |
> |3x | **0.9922** | **0.8889** | **0.5911** | **0.5853** | **0.8861** | *0.5160* | *0.5851* | *0.4869* | **0.2289** | *0.2341* | 0.2483  |
> |4x | 0.9810 | 0.6667 | *0.5834* | *0.5791* | 0.8441 | 0.4177 | 0.4977 | 0.4680 | 0.2277 | **0.2393** | 0.2470  |
>
> Among all configurations, the 3× sampling strategy consistently achieves the strongest overall performance across most metrics, demonstrating that an appropriately biased multi-task sampling schedule can effectively enhance generative capability without increasing the training budget.
>
> In contrast, sampling ratios that allocate insufficient training budget to the basic generation tasks (e.g., 1× or 2×) lead to under-optimized T2V performance. In these settings, the T2V task does not receive enough updates to fully benefit from the complementary supervision provided by other tasks.
>
> Conversely, overemphasizing the basic tasks (e.g., 4×) weakens the regularization effect brought by the editing tasks. This reduces multi-task synergy and results in performance degradation across several metrics.
>
> As an extreme case, assigning zero probability to all other tasks degenerates the training back to a pure T2V paradigm, which—as demonstrated in Table 6 of the main paper—performs worse than our mixed MfM training framework. This further validates that the improvements are not solely due to the basic tasks, but arise from the interaction among diverse tasks under a well-balanced sampling strategy.

---

> > ### Comment · Reviewer_78Yu · 2025-11-23
> >
> > Thank you for your response. I will maintain my current recommendation as positive.

---

> > > ### Author Response · Authors · 2025-11-23
> > >
> > > We sincerely thank this reviewer for the positive feedback!
> > >
> > > Best regards,
> > >
> > > Authors of paper \#2736

---

### Official Review · Reviewer_5pwg · 2025-11-03

**Soundness:** 3
**Presentation:** 3
**Contribution:** 2
**Rating:** 6
**Confidence:** 4

**Summary:**

This paper proposes a unified training framework, "Many-for-Many" (MfM), which aims to train a single deep model from scratch to support over 10 different video and image generation and manipulation tasks (e.g., T2V, I2V, video inpainting, VSR, etc.). The core of the method is a lightweight adapter designed to unify the diverse conditions from different tasks (like pixel data, depth maps, and masks) into a standardized representation. The authors also introduce depth maps as an additional 3D condition to enhance the model's 3D spatial perception and employ a joint image-video progressive training strategy. The paper's main argument is that this multi-task joint training not only enables a single model to perform multiple functions but also actually improves the performance of core video generation tasks by leveraging complementary supervisory signals from different tasks. Experimental results show that their 8B-parameter single model achieves a better average rank on both T2V and I2V tasks on the VBench benchmark compared to existing SOTA specialized models.

**Strengths:**

1. Strong Evidence of Multi-Task Synergy: The paper's strongest point is Table 6 . It clearly demonstrates the value of multi-task training. For example, the FLF2V and FLC2V tasks improve the "Dynamic" metric , while VINP and VOUTP boost semantic metrics. This shows the MfM framework is indeed learning a more robust and generalizable video representation.
2. SOTA Performance with a Single Model: Using the same 8B model, the method achieves the best "Average Rank" on both VBench-T2V and VBench-I2V. This is a very impressive result, considering its competitors (like Wan2.1, Hunyuan) are highly optimized industry models.
3. Framework Simplicity: The proposed lightweight adapter  is a clean and effective method. It processes all 2D/3D conditions uniformly and integrates them via simple addition into the DiT blocks, offering good scalability.
4. Potential Data Efficiency: While the total training data (120M videos  + 160M images ) is still massive, the video data usage is an order of magnitude less than competitors like Wan2.1 (1.5B videos) or StepVideo (2B videos). This suggests the framework may have higher data efficiency.

**Weaknesses:**

1. Limited Novelty of Components: This is my main concern. While the MfM framework and its training results are novel, the architectural components are largely a combination of existing work. The model backbone is a DiT , the training technique is Flow Matching (RF) , and the stabilization technique is QK-Norm —a combination very similar to recent work (e.g., SD3). The proposed "adapter"  also appears to be just a few convolutional layers. This makes the paper feel more like an excellent engineering and training-strategy victory rather than a paper proposing fundamental architectural innovation.
2. Lack of Analysis on Key Hyperparameter (Task Sampling Rate): The paper mentions that during training, the sampling probability for T2I, T2V, and I2V was tripled. This is clearly a critical hyperparameter for balancing the model's capabilities, yet there is no ablation study or sensitivity analysis for it. How was this 3x ratio chosen? Is this choice optimal?

**Questions:**

Please referring Weakness.

---

> ### Author Response · Authors · 2025-11-20
>
> We sincerely thank this reviewer for the recognition that our mixed training framework can achieve SOTA performance through multi-task synergy and high data efficiency. We also thank this reviewer for the insightful questions about our component design and task sampling strategy. Our itemized responses to all of this reviewer's comments can be found as follows.
>
> **Weak 1: Limited Novelty of Components**
>
> We acknowledge this reviewer’s comments that our backbone (DiT), training objective (RF/Flow Matching) and stabilization strategy (QK-Norm) are components already used in recent diffusion transformer models. However, we would like to clarify that the main goal of this work is not to introduce new architectural components; instead, we aim to demonstrate an interesting and important question: whether we can train a unified many-for-many video generation framework, which can not only support 10+ heterogeneous generation and editing tasks, but also achieve competitive performance using limited training data (40M HQ and 80M relatively LQ videos and 160M images, compared with billions of data used in other models such as Wanx, Hunyuan) and computing budget.
>
> As shown in Table 6 of the main paper, incorporating more editing tasks consistently enhances the model’s T2V generation quality from multiple perspectives. This reveals a key insight of our work: multi-task regularization and shared conditioning space lead to stronger generalization, enabling our model to match or surpass models trained on much larger datasets.
>
> Meanwhile, our adapter is deliberately designed to be lightweight. Considering that the DiT backbone is already computationally intensive, we design a 5-layer convolutional adapter that adds almost no additional FLOPs, introduces no extra attention tokens, and is general to different tasks. This design is a carefully engineered architectural choice that enables effective multi-task conditioning without increasing training or inference cost. We actually explored two alternative designs in our early experiments: (1) concatenating condition tokens along the sequence dimension, and (2) concatenating along the channel dimension.
> Both approaches substantially increased training cost—either by doubling the effective token length (condition tokens = video tokens) or by significantly expanding the input projection parameters and memory footprint as shown in the table below:
>
> | Method             | Adapter FLops | DiT Flops | Total Flops|
> |--------------------|---------|---------|--------|
> |Baseline w/o condition | 0 TFlops | 400.69TFlops | 400.69 TFlops |
> |Condition by Addition | 2.24 TFlops | 400.69 TFlops | 402.93 TFlops |
> |Condition by Channel Concat | 2.24 TFlops | 401.53 TFlops | 403.77 TFLops |
> |Condition by Sequence Concat | 2.24 TFlops | 799.09 TFlops | 801.33 TFLops |
>
> However, the empirical improvements were minimal. This led us to adopt the current additive-injection adapter design as the most effective and scalable solution.
>
> Nonetheless, we agree that a more fine-grained architecture and task-aware conditioning mechanisms may further improve performance, which is a very good direction for future work.
>
> We sincerely hope that our above explanations can address this reviewer's concern about the novelty of our work.
>
>
> **Weak 2: Ablation Study on Sampling Probability**
>
> We determined the final task ratio based on both qualitative analysis and quantitative evidence. Qualitatively, T2V, I2V, and T2I are fundamentally more difficult to optimize than other tasks because they lack strong conditioning signals. This makes them more prone to under-optimization in a mixed-task training setting. Motivated by this observation, we systematically explored how the sampling ratio of these tasks affects the model’s generative performance.
>
> As discussed in our response to weak 2 of Reviewer 78Yu, under the same training budget, alternative sampling ratios either leave the core generation tasks under-optimized or weaken the regularization benefits provided by other tasks. Among the configurations we tested, applying a 3$\times$ sampling radio to these three core generation tasks achieves the best overall performance, providing a practical and effective balance between training stability, optimization difficulty, and multi-task synergy.

---

> > ### Comment · Reviewer_5pwg · 2025-11-20
> >
> > Thanks for your responds, i will update rating if necessary.

---

> > > ### Author Response · Authors · 2025-11-24
> > >
> > > We sincerely thank this reviewer for thoughtful feedback!
> > >
> > > Best regards,
> > >
> > > Authors of paper #2736

---

> > > > ### Comment · Reviewer_5pwg · 2025-11-26
> > > >
> > > > Thank you for your response. I will maintain my current recommendation.

---

### Author Response · Authors · 2025-11-29

First of all, we sincerely thank ACs and reviewers for their time and effort. We have tried our best to address the concerns raised.

We would also like to clarify that, following our constructive discussions, all reviewers had set their scores to 6, 6, 6, and 6 as of November 24. Please refer to our detailed comments for further details. We truly appreciate everyone’s insightful feedback once again.

---

### Author Response · Authors · 2025-12-02

Dear ACs,

Following the PC's recent guidance, we are providing this brief summary to facilitate your assessment on our paper. We understand the difficult circumstances and deeply appreciate your effort in getting into this process, and we hope that our summary can be helpful for you to make a decision on our submission.

First, regarding our main contribution, as stated in the rebuttal, our work demonstrates that a simple framework, trained with limited data, can successfully support multiple generation tasks while achieving competitive text-to-video (T2V) and image-to-video (I2V) performance. Although our adapter design is simple, it effectively injects conditioning information with negligible computational overhead.

Second, regarding the training sampling rate, we clarified our motivation from two perspectives: 1) basic generation tasks are inherently more difficult to optimize so that we allocate more training budget on these tasks; and 2) our additional experiments show that using a 3× sampling probability leads to better model performance.

Finally, we addressed the remaining concerns of the reviewers through: 1) a detailed explanation of our data collection process (dataset concern); 2) additional ablations on task interactions (multi-task learning concern); 3) expanded evaluations across more video benchmarks (limited evaluation concern); 4) additional ablation study on Q-K Normalization and 3D RoPE (training stability concern).

During the rebuttal phase, all reviewers reached a consistent positive recommendation. In particular, **reviewer syLa raised his/her score from 4 to 6, resulting in a final rating of 6, 6, 6, 6**.

We also wish to explicitly clarify that **we received the final ratings on November 24, 3 days before the OpenReview incident on November 27**. We solemnly state that we did not—and would never—attempt to use the leaked information to harass any community member or manipulate the review process in any way.

Sincerely yours,

Authors of paper \#2736

---

### Meta-Review · Area_Chair_7HqP · 2026-01-06

**Summary:**

The paper is an example of system/recipe innovation rather than component innovation. The authors successfully defended their approach by demonstrating that their unified training strategy yields high data efficiency and versatile performance, which outweighs the lack of novel architectural modules. The "many-for-many" idea is an interesting direction. It would be helpful for other researchers in this community.

**Reviewer Concerns:**

The main concern is the novelty. Several reviewers criticized about the novelty/contribution of this paper. The authors are basically building a new training paradigm by using existing recipes.

Another major concern is the evaluation/ablation. The authors provided additional experiments.

**Reviewer Scores:**

The paper received mixed borderline reviews initially. I believe the authors answered the questions very well. After the rebuttal, I would hope they can maintain or increase the score a little bit, which makes it a positive paper.

---

### Decision · Program_Chairs · 2026-01-26

Accept (Poster)